

# A Neural Network Aerosol Typing Algorithm Based on Lidar Data

Doina Nicolae[1], Jeni Vasilescu[1], Camelia Talianu[1,2], Ioannis Binietoglou[1], Victor Nicolae[1,3],
Simona Andrei[1], and Bogdan Antonescu[1]

[1]National Institute of R&D for Optoelectronics, 409 Atomiştilor Str., Măgurele, Ilfov, România
[2]Institute of Meteorology, University of Natural Resources and Life Sciences, 33 Gregor-Mendel Str., 1180,Vienna, Austria
[3]Faculty of Physics, University of Bucharest, Atomiştilor 405, Măgurele, Ilfov, Romania

**Correspondence:** Doina Nicolae (nnicol@inoe.ro)

**Abstract.** Atmospheric aerosols play a crucial role in the earth system, but their role is not completely understood, partly because of the large variability in their properties resulting from a large number of possible aerosol sources. Recently developed techniques were able to retrieve the height distributions of optical and microphysical properties of fine-mode and coarse-mode particles, providing the types of the aerosols depicted. One such technique for aerosol typing is based on artificial neural

networks (ANNs). In this article, a Neural Network Aerosol Typing Algorithm Based on Lidar Data (NATALI) was developed to estimate the most probable aerosol type from a set of multispectral data. The algorithm has been adjusted for running on the EARLINET $3\beta + 2\alpha$ (+1$\delta$) profiles. The NATALI algorithm is based on the ability of specialized ANNs to resolve the overlapping values of the intensive optical parameters calculated for each identified layer in the multiwavelength Raman lidar profiles. The ANNs were trained using synthetic data, for which a new aerosol model was developed. Two parallel typing

schemes were implemented in order to accommodate datasets containing or not the measured linear particle depolarization ratios (LPDR): a) identification of mixtures from 14 aerosol mixtures (high-resolution typing) if the LPDR is available in the input data files, and b) identification of 5 predominant aerosol types (low-resolution typing) if the LPDR is not provided. For each scheme, three ANNs were run simultaneously, and a voting procedure selects the most probable answer. The whole algorithm has been integrated into a Python code. The main issue with the approached used in NATALI is that the results

are strongly dependent on the input data, and thus the outputs should be understood accordingly. The algorithm has side-applications, for example, to test the quality of the optical data and identify incorrect calibration or incorrect cloud screening. Blind tests on EARLINET data samples showed the capability of this tool to retrieve the aerosol type from a large variety of data, with different quality and physical content.

## 1 Introduction

Aerosols represent an important component of the Earth system with a significant impact on climate (e.g., Seinfeld et al., 2016), weather (e.g., Fan et al., 2016; Gayatri et al., 2017; Marinescu et al., 2017), air quality (e.g., Fuzzi et al., 2015), biogeochemical cycles (e.g., Mahowald, 2011; Mahowald et al., 2017), and health (e.g., Trippetta et al., 2016). A wide variety of aerosols are present in the atmosphere at any time originating from multiple natural (e.g., mineral dust, sea spray, biogenic emissions, volcanic eruptions) and anthropogenic sources (e.g., traffic, industrial activities, biomass burning) and having a



large variability in space and time (e.g., Calvo et al., 2013, , and their references therein). This large variety and variability of the aerosols results in uncertainties of their impact. For example, aerosols can influence the microphysical properties of clouds and hence can have an impact on the energy balance, precipitation, and the hydrological cycle.

Aerosols have different scattering and absorption properties depending on their origin, with the largest radiative contribution
coming from aerosols with radii between 0.1–1 $\mu$m (Satheesh and Krishna, 2005). Seinfeld et al. (2016) indicated that the uncertainties of the radiative forcing associated with the aerosol-cloud interactions have not changed over the last four IPCC cycles. Understanding the aerosol sources can reduce the uncertainties of their impact. Detailed knowledge of the aerosol sources can also be used to attribute their role to specific processes, evaluate aerosol models, and design better evidence-based air quality regulations.

Global and local properties of atmospheric aerosols have been extensively observed and measured using both space-borne and ground-based instruments, especially during the last decade. Satellite remote sensing observations have been exploited to characterize aerosol layers and to assess parameterizations for regional and global models (e.g., Amiridis et al., 2010). Global networks of sun/sky radiometers, such as Aerosol Robotic Network (AERONET, Holben et al., 1998) measure the spectral aerosol optical depth (AOD) (e.g., Dubovik et al., 2002; Cattrall et al., 2005; Hamill et al., 2016). The magnitude of the AOD
together with the Ångström exponent (i.e., the AOD dependence on the wavelength) can be used to infer the aerosol type although information about the source are required (e.g., Boselli et al., 2012; Giles et al., 2012). Also, the measurements are averaged over the entire atmospheric column and cannot provide information regarding the vertical distribution of particles.

Active remote sensing instruments, such as lidars, have been used to distinguish between different aerosol types (Müller et al., 2005, 2007; Groß et al., 2013; Nemuc et al., 2013; Samaras et al., 2015; Mărmureanu et al., 2016, 2017) by providing
vertical profiles of aerosol optical properties and also have been used to understand the three-dimensional structure and variability in time of the aerosol field (e.g., Freudenthaler et al., 2009; Ansmann et al., 2010; Mattis et al., 2010; Gasteiger et al., 2011a, b). Even if detailed studies of aerosol optical properties have been conducted (e.g., Brock et al., 2016a, b; Palacios-Peña et al., 2018) there are no straightforward links between the optical properties and the aerosol sources given that atmospheric aerosol occurs as a mixture of types (e.g., David et al., 2013) and thus difficult to characterize.

Recent advances in atmospheric aerosols measurements have helped address some of these issues, in particular, to separate different types of aerosols and their mixtures. For example, Burton et al. (2012) analyzed lidar measurements of aerosol parameters (i.e., lidar ration, depolarization, backscatter color ratio, spectral depolarization ratio) collected by the NASA Langley Research Center airborne High Spectral Resolution Lidar (HSRL, Hair et al., 2008) collected during measurements campaigns over North America. They showed that these parameters vary with location and with the aerosol type and thus can help
to distinguish between different types of aerosols (e.g., HSRL measurements indicated lidar ratio can be used to discriminate between ice and dust and spectral partial depolarization to discriminate between urban and biomass burning aerosols). Another important advancement in the remote sensing of aerosols was the development of ground-based lidar networks, which provide quality assured optical profiles on a large temporal and spatial scale. One such network is the European Aerosol Research Lidar Network (EARLINET) (Pappalardo et al., 2014) established in 2000 with the goal of developing a continental database of the
temporal and spatial distribution of aerosols. The EARLINET data are not only relevant for climatological studies, but also for





special events, with strong aerosol influence, like Saharan dust outbreaks, forest-fire smoke plumes transported over large areas, photochemical smog, and volcano eruptions (Tesche et al., 2009b; Mona et al., 2012; Nicolae et al., 2013; Tesche et al., 2011). Recent efforts have focused on making complementary use of different instruments such as lidar and sun or sky photometry at combined EARLINET and AERONET stations (e.g., Ansmann et al., 2002, 2010; Müller et al., 2010; Alados-Arboledas et al.,

2011; Mamouri et al., 2012; Granados-Muñoz et al., 2016; Perone et al., 2016). Several other approaches have been developed by using the combination of ground based measurements with airbore HSRLs lidars and satellite data (e.g., Liu et al., 2002; Tesche et al., 2009b; Omar et al., 2009; Kahn et al., 2010; Burton et al., 2012, 2013; Groß et al., 2013; Burton et al., 2014, 2015; Kahn and Gaitley, 2015; Papagiannopoulos et al., 2016).

All these studies have revealed the existence of a wide variety of aerosols, difficult to be classified due to a series of draw-
10 backs (e.g., many aerosol types have similar optical properties). An important issue about aerosols classification is the difficulty in correlating their optical properties to their sources. In reality, atmospheric aerosols are mixtures from many sources, and data about pure aerosol types are sparse and hard to obtain. To address these issues, systematic measurements and intensive measurements campaigns have been performed using different methods for aerosol typing (e.g. Tesche et al., 2009a; Burton et al., 2014), and complementary information such as trajectory and dispersion models analysis to estimate aerosols origin
(e.g. Stohl et al., 2003; Stein et al., 2016). Since 2000, EARLINET network has systematically measured the properties of aerosols from different sources over Europe. Intense campaigns, like ACE-Asia (Asian Pacific Regional Aerosol Characterization Experiment, Murayama et al., 2003), SAMUM-1 (Tesche et al., 2009b, Saharan Mineral Dust Experiment, Morocco), SAMUM-2 (Saharan Mineral Dust Experiment, n Cape Verde Groß et al., 2011), SALTRACE (Groß et al., 2015, Saharan Aerosol Long-range Transport and Aerosol-Cloud interaction Experiment,), ChArMEx/EMEP (Chemistry-Aerosol Mediter-
ranean Experiment, Granados-Muñoz et al., 2016) have helped in understanding the optical properties of pure types (e.g., dust and their mixtures) or anthropogenic aerosols from industrial areas. Furthermore, recent events, like the eruptions of Eyjaf-jallajökull in 2010 and Grimsvötn in 2011 offered a rare opportunity to perform studies on the optical properties of volcanic aerosols (e.g. Sicard et al., 2012; Mona et al., 2012; Tesche et al., 2012).

The multitude of instruments and retrievals resulted in an increasing amount of data on aerosols to be processed and clas-
25 sified. One possible way of processing large amounts of data with the aim of distinguishing, for example, between different aerosols types, is to exploit artificial neural networks (ANNs). Starting from the premise that the best way to distinguish between certain things and situations (e.g., image recognition, speech recognition, medical diagnosis) is the human experience based on learning and education, a mathematical method (i.e., Artificial Neural Networks, ANNs) was developed to solve problems in the same way that a human brain might. An ANN represents a mathematical projection of the brain in which the
30 information propagates as a neural influx and it is analyzed. The ANNs can contain tens to hundreds of neurons divided into multiple layers depending on the data to be classified. The output of the first layer of neurons representing the input to the next layer. The data for analysis must be constrained to a pattern and the ANNs needs to learn to identify this pattern. During the learning process, some weights of the connections between neurons are established. Learning in the case ANNs means to change these weights each time when training data are presented to the network. The change is based on the amount of error
in the output compared to the expected result.



Over the last decades, ANNs have been used amongst other applications, particularly for remote sensing applications such as radars (e.g., Orlandini and Morlini, 2000), microwave radiometers (e.g. Roberts et al., 2010), satellite retrievals (e.g., Ali et al., 2012), multi-angle spectropolarimeters (e.g. Di Noia et al., 2015), nephelometers (e.g. Berdnik and Loikov, 2016) or multiple sources datasets (Gupta and Christopher, 2009; Taylor et al., 2014). In this article, an algorithm for aerosol typing was

5 developed. The algorithm relies on a set of ANNs which are trained to recognize the aerosol type based on typical lidar data products from EARLINET, i.e., three backscatter coefficients ($\beta$) at 1064, 532 and 355 nm, two extinction coefficients ($\alpha$) at 532 and 355 nm, and, optional, one linear particle depolarization ($\delta$) at 532 nm. To distinguish between different aerosol types and their mixtures, the optical data presented to the ANNs have to be characteristic (i.e., to be independent on the density of the particles). Therefore the $3\beta+2\alpha$ ($+1\delta$) lidar data are first used to compute the intensive properties such as Ångström exponent

(AE), color ratios (CR), color indexes (CI), and lidar ratios (LR). The ability of the ANNs to retrieve the aerosol type depends strongly on the physical content and the uncertainty of the optical inputs, as well as on the structure of the ANN and the training process including the extent of the dataset used for this purpose. To create a consistent picture of the aerosol types, an aerosol model representing the optical properties of aerosol was developed. This model is capable of to reproduce the observed variable and thus can be used to construct a representative and statistically relevant synthetic databases. This synthetic dataset

is needed because of the lack of observational datasets that are statistically relevant, well-characterized, and representative for the whole spectrum of the aerosol types. Here the aerosol model was used to simulate a large number of lidar measurements (i.e., synthetic dataset) which were then used as input data to training the ANNs. The output data from ANNs consists in the most probable aerosol type within the identified layers.

This article is organized as follows. The aerosol model that was used to generate the synthetic dataset of lidar measurements
is described in section 2.1. The synthetic dataset is then used as an input for the ANNs, the core of the aerosol typing algorithm, presented in section 2.2. Sections 2.3 and 2.4 describe the neural network aerosol typing algorithm based on lidar data (NATALI). The comparison between the aerosol model output and the lidar measurements from the previous studies are discussed in section 3.1. Section 3.2 describes the performance of the ANNs. The comparison between the EARLINET-CALIPSO classification and NATALI is presented in section 3.3. Finally, section 4 summarizes this article.

## 2 Methodology

### 2.1 The aerosol model

An aerosol model was developed in this article to calculate the optical properties of pure aerosols, those generated by a single source (e.g. dust produced by the deserts, marine particles produces by the oceans). In this article, six classes of pure aerosols are considered: continental, continental polluted, dust, marine, smoke, and volcanic (Table 1). The aerosol model combines

the Global Aerosol Data Set (GADS, Koepke et al., 1997) along with the T-Matrix numerical method (Waterman, 1971; Mishchenko et al., 1996) to compute iteratively the intensive optical properties of each aerosol type. The chemical composition of each aerosol type was varied in certain limits, in order to reproduce as much as possible the large variety of particles present in the atmosphere. The synthetic database developed using the aerosol model is built for 350, 550, and 1000 nm





sounding wavelengths. These wavelengths were selected from the 61 wavelengths (0.25–40 $\mu$m) of OPAC (Optical Properties of Aerosols and Clouds) software package (Hess et al., 1998) for which the microphysical characteristics of the aerosols are available from GADS. The selected wavelengths are then re-scaled to the usual lidar wavelengths (i.e., 355, 532 and 1064 nm) using an average Ångström exponent equal to one. This was considered a valid assumption for all aerosol types, taking into

account the small difference between the lidar and the model wavelengths. If required, the aerosol model can be extended for other wavelengths.

Each pure aerosol type is built as an internal mixture of basic components which do not interact physically or chemically, having different mixing ratios. The basic components are picked up from OPAC: water soluble, insoluble, soot, mineral (nucleation, accumulation, coarse), sulphates, sea salt (accumulation, coarse). The GADS database is used for the microphysical

properties of each component (Koepke et al., 1997). However, with the current values of the complex refractive index of soot in GADS values greater than 1.2 for the Ångström exponent (550/350 nm) cannot be achieved for smoke and continental polluted types. Based on the findings of Schnaiter et al. (2003) and Henriksen et al. (2007), a typical value of 1.41 was considered for the real part of the refractive index, instead of 1.75 as it is currently in GADS.

To simulate the aerosols anizotropy, in the aerosol model the particles were considered as spheroids with different axis ratios

(Table 2). Dust and volcanic aerosols were considered prolates (i.e., axis ratio <1). Also, the proportion of soot was increased to counterbalance for the low hematite (iron oxide) content, consistent with Dubovik et al. (2002) and Gasteiger et al. (2011b).

Starting from the microphysical properties of each component, the microphysical properties of the pure aerosol were calculated by varying in certain limits one component (the critical component), while the total mixture is normalized to one (Table 2). The mixing ratio of the aerosol components is given by

$$\mu_j = \frac{N_j}{N_t}; \;\; j = \overline{1, NC} \tag{1}$$

where $NC$ represents the number of components, $N_t$ is the total number of particles, $N_j$ is the number of particles for component $j$, and the boundary condition is given by

$$\sum_j \mu_j = 1; \;\; j = \overline{1, NC} \tag{2}$$

Next, for each wavelength selected in the aerosol model, the real and the imaginary parts of the complex refractive index, were

determined with the Lorentz-Lorentz model

$$\frac{m_p^2 - 1}{m_p^2 + 2} = \sum_j \mu_j \cdot \frac{m_j^2 - 1}{m_j^2 + 2}; \;\; j = \overline{1, NC} \tag{3}$$





where $m_p$ and $m_j$ represent the complex refractive index for the particle and for the $j$ component of the particle, respectively. The aerosol particle radius is calculated with the following equation

$$r_p = \sqrt[3]{\sum_j \mu_j \cdot \left(r_j^{mod}\right)^3}; \ \ j = \overline{1, NC} \tag{4}$$

where $r_j^{mod}$ is the radius of the component $j$ with respect to humidity. Finally, the size distribution for the aerosol particle

component is considered as lognormal

$$n(r) = \frac{1}{\sqrt{2 \cdot \pi} \cdot \ln(\sigma_p) \cdot r} \cdot e^{-\frac{\ln(r) - \ln(r_p)}{\sqrt{2} \cdot \ln(\sigma)}}; \sigma_p = \sum_j \mu_j \cdot \sigma_j, \ \ j = \overline{1, NC} \tag{5}$$

The integration domain, for which the effective radius, extinction coefficient and scattering coefficient are calculated, covers medium size particles whose radius is between 0.1–5.0 $\mu$m that contribute to the scattering and extinction of light. The radius could not be increased further because of the computing and model limitations (i.e., the code used for the calculation of the

optical parameters for spheroids does not achieve the convergence for non-spherical particles). However, we do not consider this limitation as being critical for the lidar sounding wavelengths (i.e., near UV up to near IR).

Starting from the previously calculated microphysical properties and using the T-matrix code the effective cross section for the scattering and extinction of particles as well as the scattering matrix elements (phase functions) were obtained. These parameters are further used to determine (for a single particle) the aerosol optical parameters. The extinction coefficient ($\alpha$)

determined from

$$\alpha = \int\limits_{R_{min}}^{R_{max}} C_{ext} \cdot n(r) \, dr \tag{6}$$

where $C_{ext}$ represents the extinction effective cross-section. The backscatter coefficient ($\beta$) is calculated using the following relationship

$$\beta = \int\limits_{R_{min}}^{R_{max}} C_{sca} \cdot \frac{F_{11}(180°)}{4 \cdot \pi} \cdot n(r) \, dr \tag{7}$$

where $C_{sca}$ is the scattering effective cross section and $F_{11}$ is the first element of the scattering matrix (phase function). The simple albedo scattering ($\overline{\omega}$) is yielded by the ratio between the scattering and extinction effective cross sections

$$\overline{\omega} = \frac{C_{sca}}{C_{ext}} \tag{8}$$





The lidar ratio ($LR$) is determined by the following relationship

$$LR = 4\pi \cdot \frac{C_{ext}}{C_{sca} \cdot F_{11}(180°)} \qquad (9)$$

while the particle linear depolarization ($\delta$) is calculated based on the elements of the scattering matrix

$$\delta = \frac{F_{11} - F_{22}}{F_{11} + F_{22}} \qquad (10)$$

Given these conditions, the algorithm is reiterated for each composition, wavelength and humidity value until the entire selected domain is covered. The algorithm generates the properties and mixing ratio of each component, the optical and microphysical properties of the aerosol, for each wavelength, each humidity value, and each composition.

Four classes of humidity (i. e., 50%, 70%, 80%, 90%) are considered, out of the eight classes in OPAC. The high humidity values (i.e., 95%, 98%, 99%) were excluded in order to avoid ambiguous results related to activation of the hygroscopic particles. Dry particles, those with 0% relative humidity, are also not considered because they rarely present in the atmosphere. For a better representation of the particle growth, the OPAC humidity classes were linearly interpolated with a 1% step for pure types and 5% for mixed types and linearly extrapolated down to 40%. Thus, we considered that within 40%–90% range the hygroscopic growth is linear for all pure aerosols included in the model.

Even considering a certain variation of the aerosol composition and of humidity, the simulated optical parameters are not covering the whole range of measured values. This is because measurements have associated various uncertainties (i.e., systematic and statistical errors) either due to the instrument (e.g., biases, linearity, calibration) or due to the data treatment (e.g., algorithms applied to correct or average raw signals, algorithms used to calculate data products). To reproduce a close as possible the lidar measurements, we considered for each parameter an associated relative error and computed the possible range based on the initially simulated value. Within this range, any value of the parameter is accepted. This procedure is applied both to synthetic and measured data. The values considered for each parameter are generated between ($x_{med} - \Delta x$) and ($x_{med} + \Delta x$) with a certain step, which gives the finesse of the retrieval. A compromise should be made between the finesse and computing time. In case of measured data, the absolute error is taken from the measurement file and assumed to include statistical and systematic errors. In case of synthetic data, the absolute errors ($\Delta x$) have to be assumed. Based on the values reported in the literature (e.g., Ansmann et al., 2002; Freudenthaler et al., 2009), a large uncertainty is associated with the extinction coefficient, mainly due to the derivative in the inversion algorithm. Particle depolarization is very sensitive to the calibration, both for the raw signals of the two channels and for the backscatter. Thus, the values for particle depolarization have also a significant uncertainty. The backscatter coefficient calculated from the combination of Raman-elastic channels is less sensitive. Even in case of High Spectral Resolution Lidars the extinction and the backscatter are independently calculated, the cross-talk between the Mie and the Rayleigh channels are still introducing systematic errors, larger for the extinction than for the backscatter.

The relative errors considered here are 50% for the extinction, 20% for the backscatter and 30% for the depolarization. Note that these values were assumed to be inclusive (i.e., to mimic high-precision but also moderate-precision retrieved parameters).





Although for the microphysical inversion the recommended maximum value for the uncertainty of the optical parameters is 20% (Müller et al., 1999a, b), this is not critical for aerosol classification as long as a relevant number of parameters is provided (e.g., measured lidar ratios and Ångström exponent are required).

Table 4 shows the aerosol types considered in this study: six pure aerosols, seven mixtures of two pure aerosols, and two mixtures of three pure aerosols. The mixtures were obtained by linear combination of pure aerosols properties. There were a series of challenges associated with the generation of mixtures. It was considered that mixtures of two pure types were not sufficient. For example, transcontinental transport involves at least three types of pure aerosols (e.g., transport from Africa to Europe can result in a mixture of continental, dust, and marine aerosols). Adding marine changes drastically the optical properties of the mixtures of two pure aerosols. Thus, mixtures of three aerosol types were considered, especially those containing marine type. From the total number of mixtures of two and three aerosols (i.e., 35 mixtures) only those that are most frequently observed and can still be distinguished were selected (i.e., 9 mixtures). This selection of mixtures was also a compromise between the performance of the algorithm and the number of output aerosol types.

The generated optical properties of pure aerosols and mixtures serve as a basis for the determination of the Ångström exponent ($\kappa_{ext}$) and color index ($\kappa_{bsca}$) for each wavelength combination (Fig. 1). Thus, the Ångström exponent is given by the relationship

$$\kappa_{ext} = -\frac{\ln\left(\alpha_{\lambda_1}/\alpha_{\lambda_2}\right)}{\ln\left(\lambda_1/\lambda_2\right)} \tag{11}$$

Similarly, the backscatter Ångström coefficient (color index) can be determined using the equation

$$\kappa_{bsca} = -\frac{\ln\left(\beta_{\lambda_1}/\beta_{\lambda_2}\right)}{\ln\left(\lambda_1/\lambda_2\right)} \tag{12}$$

After the calculation of the spectral parameters for pure and mixed aerosols, the synthetic data are used as an input for the Artificial Neural Networks.

## 2.2 Artificial Neural Networks

The neural networks capabilities to classify data have been widely proved in many areas of research (e.g. Jain et al., 2000). Artificial Neural Networks (ANNs) are composed of a collection of connected "nodes" (i.e., artificial neurons) that are able to transmit information between one another. Similar with a neuron in the brain, an artificial neuron that receives information is able to process the information and then to transmit it as a neural influx to all artificial neurons connected to it. A comprehensive description of the ANNs theory can be found in Bishop (2000), Picton (2000), and Nielsen (2015). ANNs can be calibrated, or "trained", for a specific purpose. Here, ANNs are trained to classify aerosols using solely the lidar intensive properties as input data, without any complementary information.

The ANNs used here to classify aerosols were developed using *NeuroSolutions* a neural network development environment. Several ANNs architectures have been explored: Multilayer Perceptron (MLP), Jordan/Elman Network (JE), Generalized Feed





Forward Network (GFF), Self-Organizing Feature Maps (SOFM), Recurrent Neural Network (RNN). Each ANN architecture contains several hidden layers and different learning rules. Each layer is composed of a vector of processing elements (e.g., TanhAxon, SigmoidAxon, LinearTanhAxon) having the same parameters and an associated learning rule and learning parameters. No significant improvement in the classification of the aerosols has been achieved for different types of processing elements on

the ANN structure. Thus, TanhAxon were subsequently used. The TanhAxon applies a bias and a hyperbolic tangent function (i.e., $\tanh$) to each neuron in the layer thus ensuring that the range of each neuron is between -1 and 1. For TanhAxon the activation function is defined as

$$f\left(x_i, w_i\right) = \tanh\left[x_i^{lin}\right] \tag{13}$$

where $x_i^{lin} = \beta x_i$ and $\beta$ represent the offset. Supervised training has been used to train all ANNs. Thus, sets of input and

10 output parameters have been being successively presented to the networks for around 1000 epochs (i.e., one forward pass and one backward pass of all the training examples) per training cycle. Backpropagation is the most common form to train ANNs with more than one hidden layer. In the case of backpropagation, the weights on input elements are changed based on their previous value and a correction term. This training algorithm has been used also for the design of the NATALI ANN, the input data being continuously presented to the ANN and the output is compared with the exact aerosol type, in order to adjust the

15 weights until the desired result is achieved. The optimal values of weights and the minimum errors have been the parameters taken into account for the testing process.

Several learning rules have been tested: *Momentum*, *Conjugate Gradient Descent*, *Step*, and *Levenberg–Marquardt*. The *Momentum* learning rule is a simple and efficient approach in comparison with a standard gradient. The learning rule provides the gradient descent with some inertia, depending on the momentum parameter, which gives the smoothness of the gradient

estimation. The momentum parameter is the same for all processing elements on a layer. The *Conjugate Gradient* learning rule was also tested. It has no parameters to be adjusted, like learning rates or momentum parameter, and is faster and more accurate with respect to the standard backpropagation. The other two rules, the *Step* rule—a type of standard gradient descent algorithms that allows the user to set a default step size for all weights within the activation component, and the *Levenberg–Marquardt* rule—which gives a numerical solution to the problem of minimizing a nonlinear function, could not be used. The *Step* rule

recognized all aerosols types after first training cycle, but his active performance was low. The *Levenberg–Marquardt* algorithm blocked after several epoch. The cross-validation and test set methods have been used to stop the learning process and to assess the performances. Cross-validation monitors the error for a set of data and stops training when this error begins to increase. After a full process of training, in our case 5–10 training cycles, a testing set of data is presented to the ANN and the network output is compared with the exact type.

Sixty-eight ANN structures have been explored, starting from the simplest (reduced number of hidden layers) to the complex ones, in order to identify the best option, keeping the minimum possible time of training and testing, and avoiding "saturation" effects. Several examples from the 68 total explored ANN architectures and their structures are presented in **Table 4**. For the selection of the ANNs, the synthetic database has been split randomly into data used to train the ANN (70% of all synthetic



datasets), data used to test ANN (20% of all synthetic datasets), and data used for validation (10% of all synthetic datasets). In the training process, datasets are presented to the ANN with the correct answer. The training is performed iteratively until the classification errors are below a threshold (in our case below 25%, Fig. 2). A finer adjustment of the weighting coefficients is done during the testing process. The last 10% of the data are presented to the ANN without the known result in order to

5 validate the optimum training process and the capability of the network to classify new data inputs.

Three basic ANNs (adjusted to accommodate all data) have been chosen as appropriate to classify in parallel the multiwavelength lidar data, for both high and low-resolution classification: Jordan-Elman with 6 respectively 8 hidden layers, and Generalized Feedforward with 10 hidden layers (Table 3). The selected types of ANNs classify the aerosols based on the response with higher confidence. The ANNs have been trained using 3500 samples for each aerosol type and successive

training sessions until the best weights are reached.

### 2.3 Typing algorithm

Following the methodology described in section 2.1 and taking into account the uncertainty of each optical parameter, we generate a "bundle" of inputs for each measured or simulated aerosol layer. Answers with low confidence are filtered out (e.g. by using a threshold of minimum 0.7 confidence). The correct answer is selected then based on a statistical approach

and considering two criteria: a) which answer has a higher confidence; b) which answer is more stable over the uncertainty range. For our aerosol typing algorithm, the inputs parameters are typical data products from EARLINET database: backscatter coefficient ($\beta$) profiles at 1064, 532 and 355 nm, extinction coefficient ($\alpha$) profiles at 532 and 355 nm, and, optional, linear particle depolarization ($\delta$) profile at 532 nm. The identification of all aerosol types is not always possible, due to the dependence on the physical content (i.e., with or without $\delta$), and do the quality of the optical data (i.e., calibration, uncertainty). For

these reasons three classification schemes, with different aerosol type resolutions, are used (Table 4). First, when particle depolarization is available and all optical parameters are provided with good quality, the typing is performed in high resolution (AH). This means that the mixtures can be resolved and the number of outputs is fourteen (i.e., pure with minimum 90%, mixtures of two, and mixtures of three pure aerosol types). Second, when particle depolarization is available and the optical parameters have a high uncertainty, the typing is performed in low resolution (AL). In this case, the number of outputs is

six (i.e., pure with maximum 30% traces of other types). Third, when the particle depolarization is not available and the typing is performed in low resolution, meaning that the mixtures cannot be resolved. In this case, the predominant aerosol type is retrieved and the number of outputs is five (pure with maximum 30% traces of other types). Note that, if only spectral parameters are provided, the Volcanic type cannot be distinguished from Dust and/or Continental polluted, and therefore excluded as output.

To increase the confidence of the aerosol typing, three ANNs (see section 2.2) were developed for each three classification schemes (i.e., AH, AL, BL). A voting procedure is considered to select the most probable answer out of the three (possibly different) individual returns. The selection is made based on the confidence level of the ANN outputs, and stability over the uncertainty range (i.e., the percentage of agreement for values between error limits).





## 2.4 Python code

The Neural Network Aerosol Typing Algorithm based on LIdar data (NATALI) developed here is built on three modules: a) an input module to prepare the inputs in the specific format of the ANNs, b) a typing module to run the ANNs and decide on the most probable aerosol type and c) an output module to save the results and logs. The input module reads the lidar files in EARLINET NetCDF format, checks for the availability of all required parameters ($\beta 1064$, $\beta 532$, $\beta 355$, $\alpha 532$, $\alpha 355$, and optionally $\delta 532$ nm), identifies the layer geometrical boundaries, and calculates the intensive optical parameters within each layer, their mean value and their associated uncertainty (Fig. 3).

Layer boundaries are calculated by applying the gradient method on the 1064 nm backscatter coefficient profile (Belegante et al., 2014). The inflexion points of the second derivative of the profile data, computed with the Savitzky-Golay filter, give the top and the bottom of the layers. Gross or fine structure of the aerosol layers is revealed by a higher or lower value of the adjustable smoothing parameter (FINESSE). Only layers with a thickness larger than 300 m are considered relevant, due to the significant signal-to-noise ratio. The intensive optical parameters and their associated uncertainties are computed for the middle part of each layer for which the signal-to-noise ratio is highest (i.e., no less than 200m mid-layer), to exclude the margins which are affected by the smoothing.

The linear particle depolarization ratio is extracted directly from the EARLINET b532 file if this file exist. For each layer, the module calculates the averages and the associated uncertainties. Several filters are applied to the data, and only layers which pass these criteria are further considered for typing:

- availability of all necessary intensive optical parameters

- values of the intensive optical parameters are between acceptable limits (Table 5)

- the relative error of each intensive optical parameter is lower than 50%

For each layer and for each intensive optical parameter, the module generates a number of values (N, adjustable) between [average-uncertainty] and [average+uncertainty]. Data are than scrambled considering that any combination has a similar probability to describe the reality. The cluster of possible combinations of intensive optical parameters is then converted into the ANN input format. The typing module runs in parallel the ANNs for each dataset representing a layer, and applies the voting procedure to identify the most probable aerosol type. In case depolarization is available, the module runs in parallel six ANNs, three for high resolution (i.e., A1H, A2H, A3H) and three for low resolution typing (i.e., A1L, A2L, A3L). The probable aerosol type is provided by the high resolution ANNs, while the predominant type is provided by the low resolution ANNs. As such, if typing in high resolution fails due the data quality, the user has still access to some information in low resolution. If the depolarization is not available, the module runs in parallel three ANNs (i.e., B1L, B2L, B3L), and returns the most probable predominant aerosol type. Compared with the low typing when depolarization is available, in this case the ANNs cannot distinguish the Volcanic type, as it overlaps completely (in all existing parameters) with dust or continental polluted type. Thus, only five predominant types are retrieved. The output module prepares and saves the files in two formats csv files and human-readable (telegrams), and writes a log. The csv files and the telegrams contain contain



- identification of the datasets for which the typing was performed

- for each identified layer

    - geometrical top and bottom

    - intensive optical parameters and associated uncertainties

    - aerosol type retrieved by each ANN, and the number of agreements

    - the most probable type selected with the voting procedure (in low and high resolution separately, if is the case)

    - type of the ANN delivering the result (i.e., 1, 2, or 3)

    - comments, generally referring to situations when optical data did not passed the quality criteria, or errors in the retrieval procedure

Additional information (e.g., run time, run parameters, network error messages) are included in the telegrams. The software structure resembles the three module approach described earlier: an input module ($nt_i nput.py$), a typing module ($nt_t yping.py$), and an output module ($nt_o utput.py$). These three modules are coordinated by the $natali.py$ script, which contains the high-level algorithm and calls the required module methods.

## 3   Results

The performances of the algorithm were tested in three steps. First, the outputs of the aerosol model were compared with the literature for the values of the intensive optical parameters for each aerosol type considered in this study (section 3.1). Second, the ANNs were selected based on their performances during the learning phase, and also by comparison with a known reference (i.e., synthetic data) (section 3.2). Third, the complete NATALI algorithm was tested by comparing the retrieved aerosol types with the EARLINET-CALIPSO classification (section 3.3).

### 3.1   Comparison of the aerosol model with the literature

Synthetic aerosol optical properties, i.e., Ångström exponent ($AE_{550\_350}$), color ratios ($CR_{550\_350}$ and, $CR_{1000\_550}$), lidar ratios ($LR_{350}$ and $LR_{550}$), and linear particle depolarization ratio at 550 nm ($DEP_{550}$) generated by our aerosol model have been compared with the measured intensive parameters for the six classes of pure aerosol. The comparison with the previous literature was only possible for pure types because the properties of mixed aerosols are computed base on a linear progression of the corresponding optical properties for two pure types. As in shown in Table 6, the synthetic data are in general in very good agreement (i.e., the range of synthetic values is between the minimum and maximum values reported in the literature) with the values reported in previous studies. Synthetic values lower than those observed are for continental rural type ($AE_{550_350}$), continental polluted ($CR_{1000_5 00}$), and dust ($CR_{1000_5 00}$). Synthetic values greater than those from the literature are for continental rural ($LR_{350}$) and volcanic ($DEP_{550}$) type.



When comparing the aerosol model with the results from the previous studies, the changes in OPAC concerning the hygroscopic growth need be considered (e.g., Zieger et al., 2013). These changes have not been implemented here, because, at the time when this study was conducted, the new OPAC hygroscopicity was available. However, we speculate that the changes in OPAC will not produce major changes in the aerosol model, considering the large uncertainties introduced in the model to simulate the observations.

Figure 4 shows a comparison between the synthetic data for pure aerosol obtained from the model and the measurements obtained by Groß et al. (2013). Airborne High Spectral Resolution Lidar (HSRL) data and in situ measurements of aerosol microphysical and optical properties were collected by Groß et al. (2013) during a series of measurements campaigns in 1998 (Lindenberg Aerosol Characterization Experiment, LACE), 2006 (The Saharan Mineral Dust Experiment, Morocco, SAMUM-1), and 2008 (The Saharan Mineral Dust Experiment, Cape Verde Islands, SAMUM-2 and European integrated project on Aerosol Cloud Climate, EUCAARI). Based on these observational datasets, Groß et al. (2013) developed a aerosol classification scheme. The scheme was capable to separate between six aerosol types and aerosol mixtures (i.e., Saharan mineral dust, Saharan dust mixtures, Canadian biomass burning aerosol, African biomass burning mixture, anthropogenic pollution aerosol, and marine aerosol). The aerosol typing based on the lidar ratio and the linear depolarization ratio at 550 nm, show, in general, a good agreement between the synthetic data and the observations from Groß et al. (2013) (Fig. 4a and d). Note that smoke/biomass burning, industrial and marine type agree well. The continental and volcanic aerosol are not represented in the measurements, therefore, cannot be compared. Dust presents lower values for depolarization for the synthetic data (Fig. 4b and e), but similar values for the lidar ratio (Fig. 4c and f). Clusters are both identifiable in the synthetic and observational data. This means that for pure aerosols the combination of extinction, backscatter and depolarization at one wavelength could be sufficient for the training ANNs.

Wandinger et al. (2016) provided a synthesis of ground-based observations of lidar ratio and particle linear depolarization at 355 nm for different aerosol types (i.e., dust, smoke, pollution, marine, aerosol, volcanic ash) and mixtures, collected during a series of measurements campaigns. Compared with this synthesis of observations, the synthetic data from the aerosol model show similar values (Fig. 5). Also, the synthetic show a wider spread because of large uncertainty accepted for the input parameters. Very high values for the linear depolarization for smoke in the Aerosol CCI could not be achieved in the aerosol model.

When the entire output of the aerosol model is considered (i.e., 14 aerosol types) there is a high overlapping between clusters, in particular for mixtures, due to the built-in uncertainty (Fig. 6a). For example, smoke and continental polluted are almost completely overlapping (Fig. 6a). Due to this overlapping, the typing is difficult, which is also shown in the various measurements from the previous studies (Table 6). One parameter, which as was shown relatively recently (e.g., Freudenthaler et al., 2009), can result in better typing of the aerosol is the particle depolarization (Fig. 6b). Particle depolarization contributes to the identification of complex mixtures and to the differentiation between mineral and volcanic particles. The main issue for particle depolarization is calibration. This was only recently addressed (e.g., McCullough et al., 2017; Belegante et al., 2018) and thus few datasets satisfy the criteria for aerosol typing. Without particle depolarization, only the low-resolution typing is possible (i.e., the predominant type in a mixture is identified).





## 3.2 ANNs performance

Figure 7 shows the overall performances of the ANNs for the high-resolution typing (i.e., A1H, A2H, A3H) and low-resolution typing (i.e., A1L, A2L, A3L). In high-resolution typing at least 70% of the aerosol types defined (i.e., 10 out of 14) should be correctly assessed in more than 75% of the cases with a confidence higher than 0.7. In low-resolution typing at least 70% of the predominant aerosol types (i.e., 4 out of 5) defined should be correctly assessed in more than 65% of the cases with a confidence higher than 0.7.

The aerosol type is recognized in more than 96% of all cases in high-resolution typing (Fig. 7a). The missed cases are, in general, due to the complete overlap between the input parameters. For example, continental smoke is classified as smoke in 22% of the missed cases (i.e., 1.9% of the total number of cases). Also, continental dust is classified as dust in 9% of the missed case (i.e., 0.3% of the total number of cases), while 33% of the missed cases (1.2% of the total number of cases) are in classified as unknown

The predominant aerosol is recognized in more than 91% of the cases in low-resolution typing. Most of the missed cases are due to the ANNs not being able to distinguish between continental and smoke, and continental polluted (i.e., 36% of the missed cases representing 3.2% of the total number of cases), and continental, smoke and marine, and continental polluted (i.e., 35% of the missed cases, 3.1% of all cases). Continental polluted and marine is sometimes identified by the ANNs as continental (i.e., 27% of all missed cases, 2.4% of the total number of cases). A3H and A3L are the most performing ANNs, however not always with a high confidence level (Fig. 7). A2H and A2L have the lowest performances, but they can help in the certain cases, for example, in recognizing continental dust aerosols. The voting procedure is not always providing the right answer, as, for example, when A3H is providing the correct typing but its confidence level is low.

The dependence of the aerosol typing on humidity shows that the performances of the ANNs are decreased with an increase in humidity, for example, for continental smoke and marine mineral for high-resolution typing (Fig. 8) and for continental and smoke, and smoke, continental and marine for low-resolution typing (Fig. 8). Pure aerosol types are recognized for all values of RH. Overall, lower performances are obtained in low-resolution typing.

## 3.3 Comparison with EARLINET-CALIPSO classification

Observational data from EARLINET-CALIPSO (Cloud-aerosol Lidar and Infrared Pathfinder Satellite Observation) database and from the database developed by the of the Romanian National Institute for Research and Development in Optoelectronics (INOE) were compared with the synthetic data obtained from the aerosol model. The EARLINET-CALIPSO database, which was developed to perform measurements at EARLINET stations in correspondence with CALIPSO overpasses (Pappalardo et al., 2010), includes a total of 718 cases and 21 aerosol and cloud types. Only 13 of these cases contained all the necessary parameters (i.e., 3 backscatters, 2 extinctions and 1 depolarization). In general, the missing parameter is the particle depolarization. To increase the number of cases, the particle depolarization was added assuming typical values for the corresponding aerosol type. The assumed values were extracted from the literature. Thus, 105 cases containing all needed parameters were




obtained. Next, cases with for which all parameters are within 20% relative error. Out of 105 cases, 63 fulfill this criteria and only 57 correspond to known aerosol types.

The INOE database contains 464 measurement sets performed with the multi-wavelength Raman depolarization Lidar (RALI, Belegante et al., 2011) between June 2012–September 2014. About 55.4% of the measurements are daytime (no extinction) and 44.6% are nighttime (including extinction). Out of these, 871 processed profiles containing backscatter, extinction and particle depolarization profiles averaged over maximum one hour. Only layers with significant loads were selected, for which all intensive parameters could be retrieved with an accuracy better than 20%. Mean values within each layer were computed, excluding the edges of the layers, where the smoothing introduces large errors due to the high gradients. For each layer, the Ångström exponent, color ratio, color index, lidar ratio, and linear particle depolarization ratio were computed. Thresholds were then used to estimate at first glance the type of aerosol which resulted in a dataset with 311 cases for which aerosol type was assessed. Out of these, 182 cases were finally selected after cross-checks with auxiliary data. Thus, lidar measurements—532 nm volume depolarization time series and 355, 532, and 1064 nm range corrected signal time-series recorded by the receiver—were used to identify aerosols layers. The identification of the aerosol source was based on 96 hours backward trajectories using HYSPLIT (Stein et al., 2016). The source was assumed to originate from the region where the trajectory is closest to the ground. This method provides us guidance for identifying possible emission sources. Rainfall was also checked on the path because of possible wet deposition. A synoptic diagnosis of the main meteorological parameters (e.g., pressure, geopotential height, temperature, humidity, wind) based on NCEP/NCAR Reanalysis (Kalnay et al., 1996). This type of analysis was used to confirm the aerosols trajectories but also to determine the type of atmospheric circulation, weather regimes and weather phenomena along the trajectories.

Figure 9 shows the comparison between the aerosol typing based on the aerosol model (synthetic data) and the EARLINET-CALIPSO and INOE database (observed data). The large spread of the measured parameters is caused by the mixtures (i.e., more than two components), incorrect calibration, or subjective estimation of the aerosol type. Also, there are very observational data which lead to apparently incomplete clusters. No conclusions can be drawn for marine aerosols, as they are not represented in the observational data. Low values are observed in the Ångström exponent for several cases of dust polluted and smoke categories, as well as low values for the 532 nm lidar ration for some of the cases of continental and continental dust indicating a small portion of marine particles. This most likely due to the fact that particles are transported over a short distance above the sea before reaching the target and thus are misclassified. The high values for the Ångström exponent for some of the marine mineral aerosols indicate a mixture with smoke. High values of the Ångström exponent, greater than 1.8 as it is measured for smoke and continental polluted, could not be simulated.

## 4 Conclusions

The NATALI algorithm is based on the ability of specialized ANNs to resolve the overlapping values of the intensive optical parameters calculated for each identified layer in the multiwavelength Raman lidar profiles. The ANNs were trained using synthetic data, for which a new aerosol model was developed. Aerosols were considered spheroids and built up using OPAC-




defined internal mixtures, with the associated microphysical properties retrieved from GADS. The intensive optical properties obtained from this model were compared to literature and found to be consistent with observations. Variability of the optical properties was achieved by considering different number mixing ratios and relative humidities. In addition, the uncertainty of the observations was included as a prerequisite hypothesis in order to match the lidar data. These requirements have added

to the complexity of the ANNs selected to make the retrieval, because of the significant overlap of the input values for the intensive optical parameters. Although the linear particle depolarization ratio is a strong parameter in separating aerosol types, the measurement technology is still at the beginning, and few lidar stations provide this parameter with an acceptable accuracy. Thus, two parallel typing schemes were developed a) a high-resolution typing scheme that allows the identification of 14 aerosol mixtures if the LPDR is available in the input data files, and b) a low-resolution typing scheme that allows the identification

of five predominant aerosol types when LPDR is not provided. For each scheme, three ANNs are run simultaneously. Then a voting procedure is applied to select the most probable answer. The ANNs were selected out of 68 structures tested, as having the best performances for the aerosol typing. The voting is based on the confidence of the retrieval for each of the three ANNs, and the stability of the retrieval over the uncertainty range. A series of tests showed that considering the variation with the relative humidity from the beginning helped in making the retrieval stable for different atmospheric conditions. Also,

considering the 50% uncertainty for the input data was realistic, making possible the retrieval of aerosol types when using medium-quality lidar data, which is currently the case for research lidar networks. Without depolarization, the retrieval is much less certain, especially for mixtures, and questionable results were flagged. Spectral characteristics of volcanic aerosols are very similar with those of mineral dust and/or continental polluted, and this type cannot be distinguished if the LPDR is not provided.

The whole algorithm has been integrated into a Python code, available as source code and executable on the NATALI website (http://natali.inoe.ro/resources.html/software). The software accommodates lidar profiles—$3\beta + 2\alpha$ (+$1\delta$)—in the EARLINET data format. The software is easy to use, and a user guide is available. However, it is important that the user understands the outputs and the limitations of the algorithm. The main issue is that results are strongly depending on the input data, and the outputs should be understood accordingly. Although the neural network is able to recognize the pattern of noisy data, such

pattern has to be present and correct, otherwise, the result of the retrieval will be incorrect. The NATALI algorithm was able to

- recognize the aerosol types (high resolution, 14 types) in more than 70% of the cases for good quality optical data (i.e., the uncertainty of the intensive optical parameters less than 20%)

- recognize the predominant aerosol types (low resolution, 6 or 5 types) in more than 70% of the cases even for medium quality optical data (i.e., the uncertainty of the intensive optical parameters less than 50%)

- provide stable responses for relative humidity up to 70%, even higher for less hygroscopic aerosols

- provide results that are comparable in high and low resolution, considering the correspondence of the types defined

Furthermore, the computing time of the algorithm is relatively short due to the optimization of the Python code. The algorithm has side-applications, for example, to test the quality of the optical data and identify incorrect calibration or incorrect cloud




screening. Blind tests on EARLINET data samples showed the capability of this tool to retrieve the aerosol type from a large variety of data, with different quality and physical content. Taking into consideration the results obtained in low-resolution typing, the algorithm has a potential to be adjusted for lidar data with a lower physical content (e.g., $1\beta + 1\alpha + 1\delta$ data). For this, new ANNs have to be developed, but the aerosol model and the general retrieval scheme can be preserved. Moreover, a

similar approach could be used for any other optical instrument (e.g., photometry) as long as the physical content of the input optical parameters is sufficiently rich.

*Author contributions.* D. Nicolae carried out the research design and developed the aerosol typing algorithm. J. Vasilescu designed the artificial neural networks and conducted the statistical analysis of the output. C. Talianu developed the aerosol model. I. Binietoglou carried out the comparison of the results with the previous research and the testing of the typing algorithm. V. Nicolae developed the code for the

aerosol typing algorithm. All the authors participated in the interpretation of the results and the writing and editing process.

*Competing interests.* The authors declare that they have no conflict of interest.

*Acknowledgements.* The work presented in this paper was performed in the frame of the project Neural network Aerosol Typing Algorithm based on LIdar data (NATALI) funded by ESA under contract 4000110671/14/I-LG. Also, this project has received funding from the European Union's Horizon 2020 research and innovation programme under grant agreement No. 654109 ACTRIS-2, grant agreement No. 692014

ECARS, and Core National Program PN2018 33N/16.03.2018 funded by the Ministry of Research and Innovation. Camelia Talianu was also supported by the Austrian Science Fund FWF, Project M 2031, Meitner-Programm.



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





**Figure 1.** The generation chain of the synthetic data.



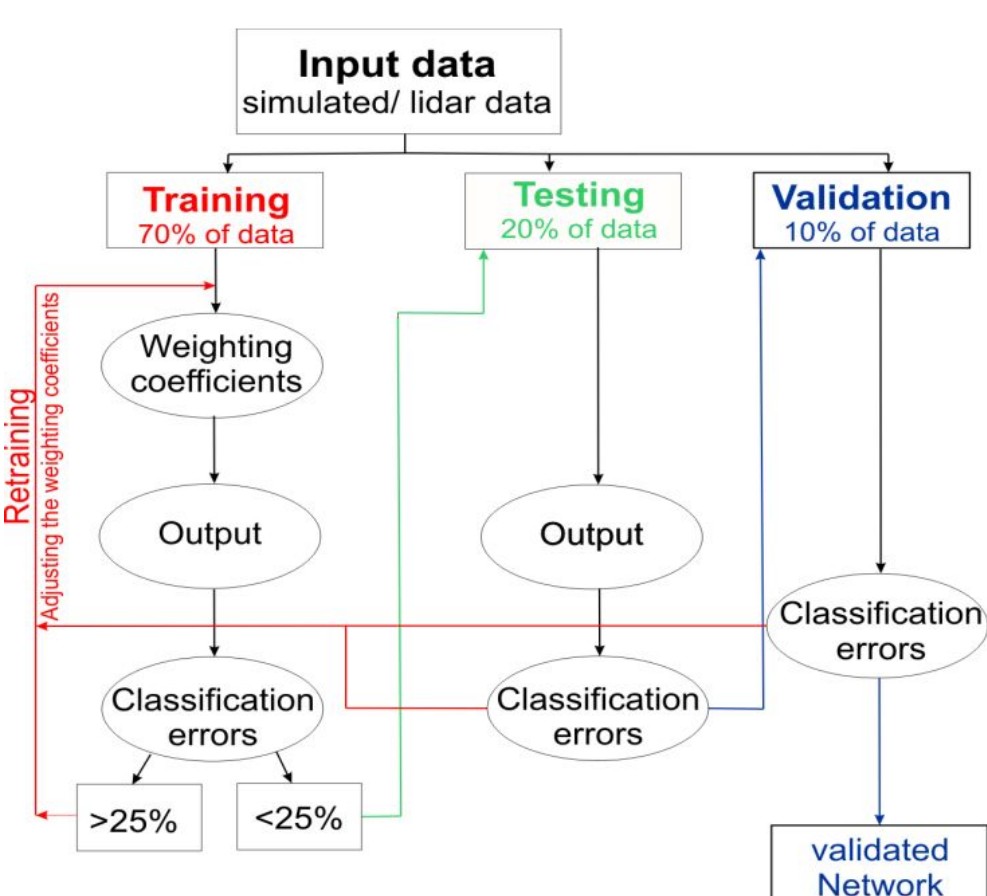

**Figure 2.** Artificial Neural Network logical scheme.





**Figure 3.** Schematics of the algorithm for aerosol typing.



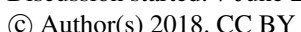

**Figure 4.** Characteristic quantities of various atmospheric aerosol types form lidar measurements (a–c, adapted from Groß et al. 2013, their Fig. 5) and from synthetic measurements (e–g). (a) and (f) Lidar ratio [$LR_5 50$ (sr)] versus linear particle depolarization [$DEP_{550}(\%)$]. (b) and (e) Linear particle depolarization versus color ration [$CR_{350\_550}$]. (c) and (f) Color ratio versus lidar ratio.





**Figure 5.** Lidar ratio versus particle linear depolarization ratio. (a) Synthesis of ground-based observations and simulation adapted from Wandinger et al. (2016) (their Fig. 1): Raman-polarization lidars POLIS(dark red dots), PollyXT measurements during SAMUM-2 at Cape Verde (Groß et al., 2011) (open squares), measurements at the EARLINET stations of Leipzig and Munich, Germany (Groß et al., 2012), in the Amazon Basin (Baars et al., 2012), and on board Polarstern over the North Atlantic (Kanitz et al., 2013). Filled stars represent simulations using the components of Aerosol CCI and variations with different refractive index and shape distribution (open stars). (b) Synthetic data from the aerosol model.





**Figure 6.** Synthetic dataset (a) color ratio versus lidar ratio, and (b) lidar ratio versus linear particle depolarization ratio.



**Figure 7.** Performances of the ANNs for (a) high-resolution typing, and (b) low-resolution typing.



**Figure 8.** Performances of the ANNs for different relative humidity values for (a) high-resolution typing, and (b) low-resolution typing.



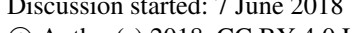



**Figure 9.** Aerosol typing based on the aerosol model (synthetic, left) and EARLINET-CALIPSO and INOE database (observed, right). (a) and (b) lidar ratio and particle depolarization (VIS). (c) and (d) Ångström exponent and particle depolarization (VIS). (e) and (f) Lidar ratio (VIS) and lidar ratio (UV).



**Table 1.** Convention names of the aerosols types.

| Aerosol type | Source | Characteristics of the particles |
|:---:|:---:|:---:|
| Continental | land surfaces | medium size, medium spherical, medium absorbing |
| Dust | desert surfaces | big, non-spherical, medium absorbing |
| Continental polluted | industrial sites | small, spherical, highly absorbing |
| Marine | sea surface | big, spherical, low absorbing |
| Smoke | vegetation fires | small, spherical, highly absorbing |
| Volcanic | volcanoes | big, non-spherical, highly absorbing |
| Mixtures | mixed | combinations of the above |



**Table 2.** Pure aerosol types and components.

| Aerosol types | Basic components types | Range variation of the critical component proportion | Axis ratio |
|---|---|---|---|
| continental | water soluble | 0.4914–0.5914 | 1.100 |
|  | insoluble | 0.0086–0.0086 |  |
|  | soot | 0.4000–0.5000 |  |
| continental polluted | water soluble | 0.1998–0.2998 | 1.040 |
|  | insoluble | 1.8e-4–1.8e-4 |  |
|  | soot | 0.6000–0.7000 |  |
|  | sulphate | 0.1000–0.1000 |  |
| smoke | water soluble | 0.3900–0.4900 | 1.150 |
|  | soot | 0.5000–0.6000 |  |
|  | sulphate | 0.0100–0.0100 |  |
| dust | water soluble | 0.1949–0.2949 | 0.870 |
|  | mineral |  |  |
|  | nucleation mode | 0.1170–0.1170 |  |
|  | accumulation mode | 0.0880–0.0880 |  |
|  | coarse mode | 0.6e-04–0.6e-04 |  |
|  | soot | 0.5000–0.6000 |  |
| marine | water soluble | 0.1652–0.1662 | 1.007 |
|  | sea salt |  |  |
|  | accumulation mode | 0.8320–0.8320 |  |
|  | coarse mode | 0.0e+00–0.1e-06 |  |
|  | insoluble | 0.5000–0.6000 |  |
| volcanic | mineral |  | 0.850 |
|  | nucleation mode | 0.0915–0.1070 |  |
|  | accumulation mode | 0.1470–0.1719 |  |
|  | coarse mode | 0.4e-04–0.5e-04 |  |
|  | sulphate | 0.0391–0.0457 |  |
|  | soot | 0.6753–0.7224 |  |



**Table 3.** Selected types of Artificial Neural Networks and their structures.

| ANN type | ANN architecture | Advantages | Disadvantages |
|---|---|---|---|
| **JE1** | 6 hidden layers<br>Momentum learning rule<br>processing elements<br> 50 for the first four layers<br> 45 for the fifth layer<br> 37 for the sixth layer<br>tanh axons<br>trained for at least 5 cycles of 1000 epochs | Good percentage of training per aerosol class. Stable performances and approximatively constant for all aerosols classes. | Slow training/time consuming. Reach the training limit rapidly. |
| **JE2** | 8 hidden layers<br>Conjugate Gradient learning rule<br>processing elements<br> 50 for the first four layers<br> 45 for the fifth layer<br> 37 for the sixth layer<br> 32 for the seventh layer<br> 28 for the eight layer<br>tanh axons<br>trained for at least 5 cycles of 1000 epochs | Good percentage of training per aerosol class. Rapid training. | Only few training cycles can be done. Limited performance improvement after training. |
| **GFF** | 10 hidden layers<br>Momentum learning rule<br>processing elements<br> 50 for the first four layers<br> 45 for the fifth layer<br> 37 for the sixth layer<br>tanh axons<br>trained for at least 5 cycles of 1000 epochs | Low error of training after two training cycles. Rapid training. | It trains efficiently only several cycles a further improvement of weights cannot be considered. Stable active performances per aerosol type overall but lower values for several classes. |



**Table 4.** Aerosol types retrieved directly and corresponding predominant types (italics).

| Aerosol types | High resolution type (AH) | Low resolution typing with depolarization (AL) | Low resolution typing without depolarization (BL) |
|---|---|---|---|
| **Continental** | Continental | Continental | Continental |
| **Continental polluted** | Continental polluted | Continental polluted | Continental polluted |
| **Smoke** | Smoke | Smoke | Smoke |
| **Dust** | Dust | Dust | Dust |
| **Marine** | Marine | Marine | Marine |
| **Volcanic** | Volcanic | Volcanic | *Dust/Continental* |
| Continental + Dust | Continental dust | *Continental/Dust* | *Continental/Dust* |
| Dust + Marine | Marine mineral | *Dust/Marine* | *Dust/Marine* |
| Volcanic + Marine | | | |
| Continental + Smoke | Continental smoke | *Continental polluted/Smoke* | *Continental polluted/Smoke* |
| Dust + Smoke | Dust polluted | *Dust/Smoke* | *Dust/Smoke* |
| Continental + Marine | Coastal | *Continental/Marine* | *Continental/Marine* |
| Continental polluted + Marine | Coastal polluted | *Continental polluted/Marine* | *Continental polluted/Marine* |
| Continental + Dust + Marine | Mixed dust | *Continental/Dust* | *Continental/Dust* |
| Continental + Smoke + Marine | Mixed smoke | *Continental polluted/Smoke* | *Continental polluted/Smoke* |





**Table 5.** Acceptable limits for the layer average intensive optical parameters.

| Intensive parameter | Minimum acceptable value | Maximum acceptable value |
|---|---|---|
| Ångström | 1±0.5 | 6±0.3 |
| Color ratio | 1±0.5 | 6±0.3 |
| Color index | 1±0.5 | 6±0.3 |
| Color index | 5±2.5 | 200±100 |
| Linear particle depolarization ratio (%) | 0±0.0 | 50±25 |



**Table 6.** Optical properties of aerosols from the synthetic dataset and from measurements.

| Aerosol type | Parameter | Synthetic | Measured | Reference |
|---|---|---|---|---|
| | $AE_{550\_350}$ | 1.17–1.29 | 2.2±0.5 | Giannakaki et al. (2010) |
| | $CR_{550\_350}$ | 1.56–2.07 | – | – |
| **Continental(rural)** | $CR_{1000\_550}$ | 1.37–1.85 | – | – |
| | $LR_{350}(sr)$ | 43–54 | 29±7 | Giannakaki et al. (2010) |
| | $LR_{550}(sr)$ | 52–53 | – | – |
| | $DEP_{550}(\%)$ | 7.23–10.7 | – | – |
| | $AE_{550\_350}$ | 1.17–1.34 | 1.4±1.0 | Giannakaki et al. (2010) |
| | | | ≈1.1–1.6 | Perone and Bulizzi (2016) |
| | $CR_{550\_350}$ | 1.34–2.29 | 0.68–0.85 | Vlăduțescu et al. (2007) |
| | $CR_{1000\_550}$ | 1.33–1.65 | 1.7–2.1 | Burton et al. (2013) |
| | | | 2.43±0.27 | Groß et al. (2013) |
| | | | 58±12 | Müller et al. (2007) |
| | $LR_{350}(sr)$ | 55–75 | 65–100 | Vlăduțescu et al. (2007) |
| | | | 56±23 | Giannakaki et al. (2010) |
| **Continental polluted/industrial** | | | 56±1 | Perone and Bulizzi (2016) |
| | | | 71±10 | Cattrall et al. (2005) |
| | | | 53±11 | Müller et al. (2007) |
| | $LR_{550}(sr)$ | 62–74 | 53–70 | Burton et al. (2013) |
| | | | 56±6 | Groß et al. (2013) |
| | | | 55±1 | Perone and Bulizzi (2016) |
| | | | 57±7 | Wang et al. (2016) |
| | | | <5 | Müller et al. (2007) |
| | $DEP_{550}(\%)$ | 2.47–4.97 | 6±1 | Groß et al. (2013) |
| | | | 3–7 | Burton et al. (2013) |



| | | | | |
|---|---|---|---|---|
| | AE$_{550\_350}$ | 1.15–1.31 | 1.0±0.5 | Müller et al. (2007) |
| | | | 1.7±0.7 | Giannakaki et al. (2010) |
| | | | 0.9±0.26 | Tesche et al. (2011) |
| | | | 0.3–0.7 | Janick et al. (2017) |
| | | | 1.0–1.5 | Stachlewska et al. (2018) |
| | CR$_{550\_350}$ | 1.90–2.59 | – | – |
| | CR$_{1000\_550}$ | 1.52–1.61 | 2.1–2.5 | Burton et al. (2013) |
| | | | 1.63±0.13 | Groß et al. (2013) |
| | | | 4.70±0.30 | |
| **Smoke** | LR$_{350}$(sr) | 56–72 | 37.9±13.1 | Müller et al. (2005) |
| | | | 46±13 | Müller et al. (2007) |
| | | | 69±17 | Giannakaki et al. (2010) |
| | | | 87±17 | Tesche et al. (2011) |
| | | | 60±20 | Janick et al. (2017) |
| | | | 55–70 | Stachlewska et al. (2018) |
| | LR$_{550}$(sr) | 81–92 | 40–80 | Wandinger et al. (2002) |
| | | | 26–87 | Müller et al. (2005) |
| | | | 53±11 | Müller et al. (2007) |
| | | | 63±7 | Noh et al. (2007) |
| | | | 60–65 | Alados-Arboledas et al. (2011) |
| | | | 79±17 | Tesche et al. (2011) |
| | | | 63±7 | Groß et al. (2013) |
| | | | 69±17 | |
| | | | 33–46 | Burton et al. (2013) |
| | | | 100±30 | Janick et al. (2017) |
| | | | 50–62 | Stachlewska et al. (2018) |
| | DEP$_{550}$(%) | 5.04–7.12 | 5–8 | Murayama et al. (2004) |
| | | | 2–3 | Müller et al. (2005) |
| | | | <2–5 | Burton et al. (2013) |
| | | | 14±2 | Groß et al. (2013) |
| | | | 7±2 | |
| | | | 3–6 | Burton et al. (2013) |
| | | | 0.93±0.01 | Burton et al. (2015) |



| | | | | |
|---|---|---|---|---|
| | | | 0.19±0.20 | Tesche et al. (2009b) |
| | | | 0.62±0.15 | Veselovskii et al. (2010) |
| | | | 1.5±1.0 | Giannakaki et al. (2010) |
| | $AE_{550\_350}$ | 0.88–0.92 | 0.06±0.21 | Groß et al. (2011) |
| | | | 0.22±0.27 | |
| | | | 0.9–0.5 | Janick et al. (2017) |
| | | | 0.0–0.3 | |
| | | | 0.01–0.18 | Fernández et al. (2018) |
| | $CR_{550\_350}$ | 1.51–1.55 | – | – |
| | $CR_{1000\_550}$ | 1.1–1.14 | 1.4–1.6 | Burton et al. (2013) |
| | | | 1.30±0.15 | Groß et al. (2013) |
| **Dust** | $LR_{350}(sr)$ | 43–46 | 55±6 | Müller et al. (2007) |
| | | | 30–80 | Papayannis et al. (2008) |
| | | | 53±7 | Tesche et al. (2009b) |
| | | | 52±18 | Giannakaki et al. (2010) |
| | | | 65±10 | Veselovskii et al. (2010) |
| | | | 58±7 | Groß et al. (2011) |
| | | | 53±5 | Groß et al. (2016) |
| | | | 42±10 | Janick et al. (2017) |
| | | | 40–55 | Fernández et al. (2018) |
| | $LR_{550}(sr)$ | 44–49 | 46±5 | Sakai et al. (2002) |
| | | | 42–55 | Liu et al. (2002) |
| | | | 55±5 | Müller et al. (2007) |
| | | | 55±7 | Tesche et al. (2009b) |
| | | | 62±9 | Veselovskii et al. (2010) |
| | | | 62±5 | Groß et al. (2011) |
| | | | 49±9 | Burton et al. (2012) |
| | | | 45–51 | Burton et al. (2013) |
| | | | 48±5 | Groß et al. (2013) |
| | | | 56±7 | Groß et al. (2015) |
| | | | 32±10 | Janick et al. (2017) |
| | | | 38–61 | Fernández et al. (2018) |





| | | | | |
|---|---|---|---|---|
| **Dust** | DEP$_{550}$(%) | 27.22–30.97 | 10–35 | Müller et al. (2007) |
| | | | 10–25 | Papayannis et al. (2008) |
| | | | 32±2 | Freudenthaler et al. (2009) |
| | | | 31–33 | Burton et al. (2013) |
| | | | 24–27 | Groß et al. (2011) |
| | | | 31±25 | Groß et al. (2013) |
| | | | 26±3 | Groß et al. (2015) |
| | | | 32.7±001 | Burton et al. (2015) |
| **Marine** | AE$_{550\_350}$ | -0.26–0.21 | – | – |
| | CR$_{550\_350}$ | 0.77–1.35 | – | – |
| | CR$_{1000\_550}$ | 0.7–2.91 | 1.3–1.6 | Burton et al. (2013) |
| | | | 1.64±0.10 | Groß et al. (2013) |
| | LR$_{350}$(sr) | 13–32 | 18±4 | Groß et al. (2011) |
| | | | 20±3 | Groß et al. (2016) |
| | LR$_{550}$(sr) | 19–25 | 28±5 | Cattrall et al. (2005) |
| | | | 23±3 | Müller et al. (2007) |
| | | | 18±2 | Groß et al. (2011) |
| | | | 15–25 | Burton et al. (2012) |
| | | | 17–27 | Burton et al. (2013) |
| | | | 18±5 | Groß et al. (2013) |
| | | | 22±5 | Groß et al. (2016) |
| | DEP$_{550}$(%) | 1.9–3.73 | 2–3 | Groß et al. (2011) |
| | | | <10 | Burton et al. (2012) |
| | | | 4–9 | Burton et al. (2013) |
| | | | 3±1 | Groß et al. (2013) |
| | | | 2±1 | Groß et al. (2016) |





| | | | | |
|---|---|---|---|---|
| **Volcanic** | $AE_{550\_350}$ | -0.21–1.07 | 0.03±0.4 | Ansmann et al. (2010) |
| | | | -0.11±0.4 | |
| | | | 0.68±0.63 | Sicard et al. (2012) |
| | $CR_{550\_350}$ | 0.82–1.29 | – | – |
| | $CR_{1000\_550}$ | 0.74–2.57 | – | – |
| | $LR_{350}(sr)$ | 50–54 | 60±5 | Ansmann et al. (2010) |
| | | | 30–60 | Mattis et al. (2010) |
| | | | 39±10 | Sicard et al. (2012) |
| | | | 60±11 | Mona et al. (2012) |
| | $LR_{550}(sr)$ | 41–49 | 60±5 | Ansmann et al. (2010) |
| | | | 30–45 | Mattis et al. (2010) |
| | | | 32±4 | Sicard et al. (2012) |
| | | | 78±12 | Mona et al. (2012) |
| | $DEP_{550}(\%)$ | 37.27–41.8 | 0.33±0.03 | Ansmann et al. (2010) |
| | | | 16±7 | Mona et al. (2012) |