# Peer review of "A Neural Network Aerosol Typing Algorithm Based on Lidar Data"

_Atmospheric Chemistry and Physics, 2018_

## Referee Comment (RC1) · Anonymous Referee #2 · 3 Jul 2018

1) page 2 line 28, delete "collected" (repetition) 2) page 3 line 1, you can add references for forest-fire smoke plumes transported over large areas 3) page 4 line 13, delete "of" after the word "capable" 4) page 7 line 17, "as close as" instead of "a close as" 5) page 11 line 33, delete the second "contain"

---

## Referee Comment (RC2) · Anonymous Referee #1 · 9 Aug 2018

The ACPD paper entitled A Neural Network Aerosol Typing Algorithm Based on Lidar Data by Nicolae et al. plays certainly a difference in the atmospheric aerosol typing research.

The proposed aerosol typing algorithm provides realistic range of results, as compared to values reported in literature in last two decades. The classification is done for pure aerosol, different component mixtures and/or predominant aerosol type. The selection of aerosol classes is scientifically relevant and detailed.

The authors made the developed code available, which should be stressed as a section related to Software Availability, after/before Acknowledgements.

I am somewhat missing discussion of the lower and upper range-limit of the typed

layers. How deep does it go into the boundary layer? What is the highest aerosol layer observed? Especially, for the validation exercise this would be relevant to know.

In Conclusions, I would expect more on the outlook.

The paper is written in an easy to read, casual manner. I made many specific comments that should be addressed to clarify the methodology. Suggestions for cosmetic changes in mathematical formulas are outlined. The Figures and Tables are in general good. I proposed some improvement. The captions of Figures and Tables are not acceptable, as they do not guide the reader.

The language seem in general fine for non-native speaker. In some sentences I suggest corrections and pointed typos.

I listed below many comment, all of which are actually only minor revisions. Therefore, I recommend this article for publishing, provided minor revisions are undertaken.
* * *
Minor revisions & comments

Throughout the text, please consider following changes: 1) remove coma after i.e. & e.g. 2) change 'ration' to 'ratio' 3) change 'humidity' to 'relative humidity' 4) change 'backscatter' to 'backscattering' 5) check color ratio vs color index - these are mixed up and the latter is not defined 6) fonts are sometimes mixed up (on purpose?) in text, tables and equations, what looks strange 7) make a decision to use the names of aerosol types either with small-latter or with Capital letter, and keep it consistent.

p.1 l.3 lidar-based techniques

l.4 remove 'depicted'

l.4 change to 'One of such techniques in based'

l.6 change to 'multispectral lidar data'

l.8 change to 'parameters, calculated'

l.10 change to '(or not)'

l.11 change to 'identification of 14'

l.13 change to 'the most probable aerosol type'

l.14 change to 'The limitation of NATALI is'

l.15-16 change to 'Additional applications of NATALI, e.g. to test'

l.16 change to 'or insufficient cloud screening is feasible.'

l.17 change to capability of NATALI to retrive

p.2 l.1 change to' (Cavalo et al., 2013, and references therein)'

l.6 change to 'IPCC reporting'

l.7 change to 'sources should reduce'

l.15 change to 'aerosol type,'

l.16 change to 'However, the measurements averaged ... column cannot'

l.18-20 move references as: 'aerosol types by providing ... properties (Muller ...), as well as to understand'

l.24 change to '(e.g. David...), thus difficult'

l.25 change to 'helped to address'

l.28 remove 'collected'

l.31 spectral particle depolarization ?

p.3 l.1 change to ',such as Saharan' and later on 'large areas and long distances'

l.2 references - add also more recent demonstration of the network capability, e.g Atmos. Chem. Phys., 17, 5931-5946, https://doi.org/10.5194/acp-17-5931-2017, 2017

l.10 change to 'Another issue in aerosol'

l.12 change to 'data of pure aerosol types are sparse.'

l.17-19 for consistency, move ref.Tesche after 'Marocco,' and ref.Gross15 after 'interaction Experiment,'

l.18 remove 'n' and add coma before ref.Gross11

l.20-21 clarify: ... of pure types (e.g. dust and their mixtures) <- so pure or mixtures?

l.24 change to 'aerosol properties to be'

l.25 an aim? and later remove ', for example,'

l.27 consider changing 'things and situations' to 'data'

l.30 change to 'The ANN contain tens'

p.4 l.4-5 change to 'In this paper, an in-house-developed ANN algorithm for aerosol typing is introduced.'

l.9 change to 'are used at first to compute'

l.10 start in new line: The ability...

l.12 coma before 'including'

l.13 change to 'of different aerosols was' later on 'capable of reproducing the observed aerosol properties, and'

l.15 change to 'needed doe to sparse observational data sets'

l.16 change to 'The aerosol model was constructed to'

l.27 remove 'in this article'

l.31 chemical composition <- give hint on which parameters

l.32 certain limits <- specify in brackets what limits you mean

l.32 remove 'as much as possible'

p.5.l.4 remove 'average'

l.14-15 change to 'In the aerosol model, particles ... ratios to simulate the aerosol anisotropy'

l. 17 microscopical properties <- give hint on which properties

l.18 change to 'in certain limits the critical component (in brackets give hint what you mean!)'

l.24 change to 'For each'

p.6 l.1 change to 'for the j components of the aerosol mixture, respectively.'

l.4 change to 'The size ... is mono-modal, log-normal and given by ..... where sigma_p, sigma_j sigma is ... and r is ... ' (note: in Eg.5 is r not r_j ?)

l.13 change to 'Use of the calculated microphysical properties with the T-matrix code (provide reference), the effective cross-section for the particle scattering C_sca and extinction C_ext, as well as ...'

l.14-15 and l.17-18 change to '... alpha is determined from Eq.6 and ... beta from Eq. 7, where F11 is...'

l.7-11 move to the end of l.20, then correct it as follows: The integration domain (R_min:R_max), for which the effective radius r?, r_p?, the extinction coefficient alpha and scattering coefficient sigma? sigma_p? are calculated, covers medium size particles with radius between ... The radius was not increased further due to computing time limitation and model design limitations (i.e. ...). However, the latter limitation is not considered critical for the range of the lidar wavelengths.

l.10 correct and start in new line: 'The single scattering albedo omega is yield as the ratio of the ....

p.7 l.5 change to 'The algorithm is iterated for each'

l.6 it is confusing -> what domain you mean here?

l.11 in brackets change to '(above 90%)'

l.10 change to 'considered as too rarely present in the ambient atmosphere.'

l.12 change to 'Thus, within ... is considered linear for '

l.15 rather: 'This is partly due to the limitations of the model itself and partly due to the various uncertainties associated to the measurement ...'

l.16 explain what you mean by linearity here.

l.17 change to 'applied in pre-processing to correct' later on change to 'as close'

l.17-21 this is not very clear, do you mean that 'To reproduce lidar measurement, for each simulated parameter an associated relative error (deltax) is assumed and the possible variable range is then computed as [x_med-deltax:x_med+deltax]', were x_med is the mean? median? value within the selected layer boundaries? But then why this is applied to syntetic data and true measurement? So you add additional error to the true data? what you mean by 'certain step' in finesse? is the deltax relative or absolute error and finally which is assumed, both?

l.23 start in new line: Based on the values ...

l.25 rather it should be: 'extinction coefficient derived with Raman method, mainly due to noisy Raman lidar signals.'

l.24,27, be specific, in brackets give uncertainties of ext, bscat and depol that you mean!

l.27-28 change to 'in the case'

l.28 change to 'Lidars, were the'

l.29 change to 'cross-talk... is still'

l.31 change to 'inclusive to mimic ...parameters.'

p.8 l. 2 add come before 'as long as'

l.5-6 remove sentence 'There was.. mixtures' and then change to 'The mixtures composed of only two pure types were considered not sufficient.'

l.10-11 change to 'of possible mixtures ... three pure aerosols (35 mixtures), only ... (9 mixtures, see Table 4).'

l.12 you mean: the time-performance of the algorithm and the minimum number of output aerosol types considered significant in atmosphere.

l.13-14 correct and change to 'the extinction Angstrom exponent (k_ext) and the backscattering Angstrom exponent (k_bsca), also referred to as color ratio'

l.21 titla of 2.2: The ANN architecture ?

l.22-26 belongs to Introduction! -> start the section with: The ANNs can be calibrated or...

l.2 change to 'elements of identical parameters (e.g. ...) with an associated ...'

l.4 start as new paragraph: No significant ...

l.5-6 unclear sentence: 'The TanhAxon ... -1 and 1.' you mean: a bias-offset B ? each neuron x_i ? in the layer omega_i ?

l.9 remove 'all'

l.9 start in the new line with: Supervised ...

l.13 change to ' training approach... ANN: the input... and the output compared with the [specify: true? or initially assumed? or pure?] aerosol type, in order...'

l.15 specify: the minimum classification errors ?

l.17 clarify: the standard gradient was not tested at all?

l.18 change to 'It provides'

l.21 change to The ConjugateGradient ... has no parameters ... (such as learning ...) and'

l.22 change to 'algorithm'

l.24 change to 'function, were found inadequate for aerosol typing purpose.'

l.25 change to 'but its active'

l.26 start as new paragraph: The cross-validation ...

l.27 please make it more specific here what you mean by 'when error begins to increase'? is it at the beginning zero? or has some initial constant value? how is it defined? assumed?

l.29 be more specific, what you mean by the exact type?

l.30-31 change to In total 68 ANN structures ... in order to compromise between the minimum ...

l.31 specify what is meant here by saturation effects?

l. 32-33 rather change to 'Examples of 6 pure, 7 double-component mixtures and 2 triple-component mixtures obtained within the 68 explored ANN ...

p.10 l.2-5 state clearly if for the training, tasting and validation the same classification error <25% is used.

l.7-8 should this not read 'the Jordan-Elman with 6 or 8 hidden laers and the Generalized Feedforward with 10 hidden layers' ?

l.9 what you mean by response with higher confidence? the classification error?

[Figure]

l.10 define the bast weights

l.11 the title of 2.3 The typing algorithm ?

l.12 should not be 'the uncertainty threshold'?

l.12-13 change to 'parameter, a bundle of imputs ... was generated.'

l.14 remove: 'then'

l.15 remove: 'and'

l.16 start at new line: For the NATALI aerosol typing algorithm, the input ...

l.17 optionally?

l.18 remove: all

l.21 good quality <- specify in brackets the uncertainty threshold

l.24 high uncertainty <- specify in brackets the uncertainty threshold range

l.25 change to 'available, the'

l.26 change to 'meaning again that the'

l.27 change to 'retrieved for 5 outputs (...), whereby if... from Dust nor Continental ..., and therefore is ...'

l.30-31 change to 'The three ANNs (Table 3) were developed ...schemes (Table 4) to increase the confidence of the aerosol typing. A voting procedure selects the most...

p.11 l.1 the title of 2.4 The NATALI Code ?

l.2 change to 'The ... developed in the Phyton programing environment is buitl on ...

l.6-7 change to 'calculates within each layer the mean intensive optical parameters (name them in brackets) with uncertainties (Fig.3).'

l.8 should be: The layer boundaries

l.8 please clarify what is used boundary layer detection: the gradient (1st derivative of signal) or as in l.9 the inflection point (2nd derivative of signal)

l.8 is the range&time resolution of 1064 signal the same as the resolution of the optical EARLINET profiles?

l.9 specify what parameters for SG-filter are used, are they applied the same for all profiles?

l.11-14 From own experience, layers of thickness < 300m can have higher SNR than layers > 300m, as SNR depends on aerosol optical depth, which even for very thin layers can also be very high! I suggest to revise the fragment as follows: The layers with thickness of more than 300 m are considered, whereby the intensive optical properties and their uncertainties are computed for the middle of each layer in the range of at least 200m thickness, to exclude the margins likely affected by the smoothing.

l.15-16 The first 2 sentences are repetitions, pls remove it, and start directly with: Several filters ... pass the following criteria ...

l.22 change to 'the input model generates an adjustable number N of values x with uncertainties deltax in the range of [x-deltax:x_deltax]'.

l.24 start in new line: The typing module ...

l.25 change to 'In the case that the depolarization ratio is'

l.28 remove 'some'

l.29-32 change to ', and returns only the ... type (Volcanic overlaps completely (...) ... polluted type and cannot be retrieved in low-resolution.'

l.32 remove as irrelevant sentence: Thus...

l.33 change to 'telegrams contain the identification of the data sets for which typing is

performedand provides for each identified layer following parameters:'

p.12 l.3-7 start each parameter with 'the'

l.10 change to 'The NATALI code'

l.10-12 check the names of the python routines there is problem with the font.

l.12 should be: The three

l.13 rather module routines/codes than methods!

l.15-18 should be Firstly, ... Secondly, ... Thirdly, ...

l.22 change to 'by the developed aerosol model'

l.25change to 'As shown in'

l.26 move the '(i.e. ...) to the end of the sentence.

l.27-29 could you comment on what could be the reason for the given discrepancies?

p.13 l.3 OPAC hygroscopicity was NOT available?

l.3-4 change to 'However, the changes in OPAC are not expected to produce ...'

l.6-8 chnage to 'In the Figure 4 comparisons ... by Gross et al. (2013) are provided. =

l.8-13 Based on the Airborne... properties measured during campaigns in ... EU-CAARI), Gross et al. (2013) developed an aerosol classification scheme for six aerosol types ...'

l.15 should be: observations at 532 nm from ... (Fig.4 a and d), especially for smoke...

l.17 should be: therefore were not compared.

l.18-20 should be: Clusters are identified both in synthetic and observational data , which means ... one wavelength can be sufficient for the ANN training.

l.18-20 I am interested if you did actually try to train the ANNs with the single wavelength extinction, backscattering and depolarization?

l.23 either add after 'measurement campaigns' comment in brackets: (listed in caption of Fig.5)' or list the campaigns in the text and leave them out in the Fig.5 caption, what will be consistent with Fig.4

l.29 it is difficult to see overlapping of smoke and continental polluted

l.29-31 I would revise this fragment as follows: Smoke and continental pollution almost completely overlap (Fig.6a), which is consisted with measurements reported in literature as in Table 6. This makes the typing challenging. The importance of particle depolarization shown relatively recently (e.g. =Freudenthaler et al., 2019), can improve the aerosol typing (Fig.6b).

l.33 should be: calibration, recently addressed by e.g. McCullough ...

l.34 should be: the depolarization ratio quality criteria.

l.34-35 rephrase this sentence to positive: However, even without particle depolarization information, the low-resolution typing can identify the aerosol predominant types in a mixture.

p.14 l.9 change 'Also,' to '; continental ...'

l.10 change 'while' to '.Note that 33% ...'

l.12 add at the end of 1st sentence: (Fig.7b).

l.16 start in new line: A3H and A3L ...

l.16 most or best performing ?

l.18-19 last sentence -> pls clarify, I do not see this from provided Figs!

l.21 change 'for example,' to 'olny'

l.21 I think there should be dust continental instead of marine mineral for high-resolution typing

l.22 I think there should be continental smoke and mixed smoke for low-resolution typing

l.23 consider adding sentence: For costal polluted, the relative humidity increase results in an increase of typing performance.

l.25 I suggest to separate the CALIPSO and INOE data sets, therefore I propose to rewrite this as: Observational data from EARLINET Data Base (www.earlinet.org), related to the CALIPSO (...) overpasses over different EARLINET observational sites, were compared with ... model. The EARLINET-CALIPSO Data base (Pappalardo et al. 2010), covers the data of 2000-2018 ??? and includes...

l.29 should be: all of the

l.31-32 change to: '... was added, assuming values reported in literature as typical for the corresponding aerosol type. This way, 105 cases ... were obtained.'

p.15 l.1-2 change to: 'The cases for which all parameters were within 20% of relative error were selected (63 cases), whereby 57 corresponded to known aerosol types.'

l.3 add sentence: 'Additionally, profiles available at the EARLINET site in Bucharest/Magurele, established by the Romanian National Institute for Research and Development of Optoelectronics (INOE), were used to increase the validation measurements sample.'

l.4-5 change to: 'About 44.6% of measurements were conducted at nighttime (thus, include the Rama-derived extinction coefficient profiles).

l.5 871 profiles or layers?

l.6 remove: profiles

l.6 instead of maximum of 1h, give range of temporal averaging (e.g. 30min-1h)

l.6 significant aerosol load ?

l.7 change 'could be' to 'were' later on should be 'higher than 20%'

l.10-11 in fect this are 311 and 182 layers?

l.11 start last sentence is new line as: The time-series of lidar measurements (532 nm vol.depol.ratio and ... range corrected signals) were used to identify the aerosol layers.

l.13 start last sentence in new line : The identification...

l.14-16 change to: 'source was assumed at the region ... was closest to the ground, providing guidance ... The rainfall along trajectory was used a indicator for likely wet deposition.'

l.16 meteo parameters or fields?

l.17-18 should be: was based on ... reanalyses (Kalnay et al., 1996) to confirm the aerosol trajectories ...

l.21-23 be specific: the mixtures of three components, incorrect calibration or inappropriate estimation of aerosol type. On the other hand the sparse observational data lead to ...

l.25 make new sentence: Low values ... lidar ratio are seen for several cases of ...

l.26 should be: This is most likely ...

l.27 coma before 'and thus'

l.29 change 'could not' to 'failed to'

l.29 consider adding information: Actually, in general the agreement of classification of the simulated data and the real observations is very good, given all limitations discussed.

p.16 l.2-4 rather unclear, especially the prerequisite hypothesis

l.6 strong or crucial?

l.7 should be: the depolarization methodology is still maturing and only a few lidar stations ...

l.15 should be: for the input data gave realistic retrieval of aerosol ...

l.21 I reckon you should repeat very valuable information in the section: Software Availability next to the Acknowledgements!

l.22 easy to use or: The NATALI software is user-friendly; a user-guide is provided.

l.23 should be: the limitations of the algorithm, i.e. the results are ... on the quality of the input data

l.27 should be: of less that

l.28 rather: of the cases for medium and high

l.33 rather: for example, it can be applied to test ... data, identify ...

l.33 could you please comment whether you did actually tested this specific QC-procedure? is there any conference paper on that you could cite?

p.17 l.2-3 comment on expected results for data sets of more than 3beta, 2 alpha and 1sigma, e.g. adding polarization at 355 and/or 1064 nm; adding lidar-derived RH profiles? How you expect this improves the results?

—-

References

p.21 l.17 should be Janicka; p.25 l.10 should be Muller, D.; p.25 l.13 remove 2x .and

—-

Equations

Eg.9 more elegant formulation: LR=4pi/omega*F11(180)

Eg. 13 please do not use beta and omega in this equation, as they both have already associated physical quantities! the current beta-bias-offset can be denoted as B. What is the omega here? as there is x_iˆlin is this not LinearThanAxon?

—-

Figures

Fig.1&2&3 improve captions, specifically add info that schemes are used for the NATALI code.

Fig.3 I suggest to change the description to: Mean intensive optical parameters within the layer, Mean uncertainty of intensive optical parameters within the layer, Scrambled seta of mean intensive optical parameters +- uncertainty

Fig.4 indicate in upper sub-figs. what wavelengths they depict; for consistency in legend of lower sub-figs give aerosol types in sequence: dust, dust polluted, smoke, continental polluted, marine

Fig.5 indicate for which wavelength is the upper figure! for comparisons the same wavelengths of the observations and synthetic data must be used. Caption is lengthy, the description of the campaigns should go to the text.

Figs.6-9 for consistency, order the aerosol types as in AH definition in Table 4 and use THE SAME COLOR SCALE for each aerosol type.

Fig.6 upper figure should be CRvsLR not ARvsDR! In caption add info that this is NATALI classification

Fig.7&8 change Marine/CC to marine! Plot confidence-levels? vote-number? in grayscale (white being 0-20%, black 80-100%). Note that low-resolution typing for 7 pure

categories is excellent. Note that high and low resolution results are better of any info on RH is available. Give more explanation on how to understand figs in the captions. Again, add info that this is for NATALI code.

Fig.9 Add info that INOE site is also EARLINET site. Add over the left column info synthetic-data and over the right one observational-data

—-

Tables

Table 1 change big to large; change right column title to Particle characteristics, for marine -> should not be non-absorbing? continental polluted and smoke are identical but smoke can be aspherical.

Table 2 what is in the 3th column, whet is this critical component proportion? is has no units? you mean axis-ratio or aspect-ratio? Is this table defined by results reported in literature?

Table 3 check font of than-axons

Table 4 specify whether aerosol types are retrieved directly by the NATALI or measurements? Note there is no italic font! why categories in Arial-font are given in low-resolution if they are not really retrieved?

Table 5 AE and CR have the same values? is the AE, CR of 6 realistic? the Rayleigy being 4 should not be the limit? Which CI is correct? Is the lower CI not LR? Please check carefully this table for typos!

Table 6 consider adding reference for Continental polluted/industrial: Raman lidar-derived AE550/350 of 1.17-1.19 is reported in Remote Sensing 2017, 9(11), 1199; doi:10.3390/rs9111199.

---

## Author Comment (AC1) · 19 Sep 2018

1) page 2 line 28, delete "collected" (repetition) Deleted from the manuscript. 2) page 3 line 1, you can add references for forest-fire smoke plumes transported over large areas A reference was added to "Vaughan, G., A. P. Draude, H. M. A. Ricketts, D. M. Schultz, M. Adam, J. Sugier, and D. P. Wareing, 2018: Transport of Canadian forest fire smoke over the UK as observed by lidar, Atmos. Chem. Phys., 18, 11375–2 11388, 10.5194/acp-18-11375-2018." 3) page 4 line 13, delete "of" after the word "capable" Deleted from the manuscript. 4) page 7 line 17, "as close as" instead of "a close as" The paragraph was replaced in the new version of the manuscript. 5) page 11 line 33, delete the second "contain" Corrected in the manuscript.

---

## Author Comment (AC2) · 19 Sep 2018

We would like to thank the reviewer for his/her positive and insightful comments on the manuscript. Below is our responses to the issues raised in the review. The comment from the reviewer and the response are separated by "–".

Minor revisions & comments

Throughout the text, please consider following changes: 1) remove coma after i.e. & e.g. 2) change 'ration' to 'ratio' 3) change 'humidity' to 'relative humidity' 4) change 'backscatter' to 'backscattering' 5) check color ratio vs color index - these are mixed up and the latter is not defined 6) fonts are sometimes mixed up (on purpose?) in text, tables and equations, what looks strange 7) make a decision to use the names of

aerosol types either with small-latter or with Capital letter, and keep it consistent.

Throughout the manuscript: 1) comma was removed after e.g. and i.e., 2) "ration" was changed to "ratio", 3) "humidity" was changed to "relative humidity", 4) we have used "backscatter" because, as shown by recent ERALINET publications in ACP, this term is the preferred one, 5) both color ratio and color index are now clearly defined in the manuscript, 6) the issues related with fonts was corrected, and 7) the aerosol types are in small caps.

p.1 l.3 lidar-based techniques

–Changed in the manuscript.

l.4 remove 'depicted'

–Removed from the manuscript.

l.4 change to 'One of such techniques in based'

–Changed in the manuscript.

l.6 change to 'multispectral lidar data'

–Changed in the manuscript.

l.8 change to 'parameters, calculated'

–Corrected in the manuscript.

l.10 change to '(or not)'

–Changed in the manuscript.

l.11 change to 'identification of 14'

–Corrected in the manuscript.

l.13 change to 'the most probable aerosol type'

[Figure]

–Changed in the manuscript.

l.14 change to 'The limitation of NATALI is'

–Changed in the manuscript.

l.15-16 change to 'Additional applications of NATALI, e.g. to test'

–Changed in the manuscript.

l.16 change to 'or insufficient cloud screening is feasible.'

–Changed in the manuscript.

l.17 change to capability of NATALI to retrieve

–Changed in the manuscript.

p.2 l.1 change to' (Cavalo et al., 2013, and references therein)'

–Corrected in the manuscript.

l.6 change to 'IPCC reporting'

–Changed to "IPCC reports".

l.7 change to 'sources should reduce'

–Changed in the manuscript.

l.15 change to 'aerosol type,'

–Corrected in the manuscript.

l.16 change to 'However, the measurements averaged ... column cannot'

–Corrected in the manuscript.

l.18-20 move references as: 'aerosol types by providing ... properties (Muller ...), as well as to understand'

–Changed in the manuscript.

l.24 change to '(e.g. David...), thus difficult'

–Changed in the manuscript.

l.25 change to 'helped to address'

–Corrected in the manuscript.

l.28 remove 'collected'

–Removed in the manuscript.

l.31 spectral particle depolarization?

–Changed to "spectral particle depolarization".

p.3 l.1 change to ',such as Saharan' and later on 'large areas and long distances'

–Changed in the manuscript.

l.2 references - add also more recent demonstration of the network capability, e.g At-mos. Chem. Phys., 17, 5931-5946, https://doi.org/10.5194/acp-17-5931-2017, 2017

–Citation added to Ortiz-Amezcua, P., Guerrero-Rascado, J. L., Granados-Muñoz, M. J., Benavent-Oltra, J. A., Böckmann, C., Samaras, S., Stachlewska, I. S., Janicka, Ł., Baars, H., Bohlmann, S., and Alados-Arboledas, L.: Microphysical characteriza-tion of long-range transported biomass burning particles from North America at three EARLINET stations, Atmos. Chem. Phys., 17, 5931-5946, https://doi.org/10.5194/acp-17-5931-2017, 2017.

l.10 change to 'Another issue in aerosol'

–Changed in the manuscript.

l.12 change to 'data of pure aerosol types are sparse.'

–Removed from the manuscript.

l.17-19 for consistency, move ref.Tesche after 'Marocco,' and ref.Gross15 after 'interaction Experiment,'

–Corrected in the manuscript.

l.18 remove 'n' and add coma before ref.Gross11

–Removed from the manuscript.

l.20-21 clarify: ... of pure types (e.g. dust and their mixtures) <- so pure or mixtures?

–Changed to "[. . .] optical properties of aerosols (pure dust and mixtures) or anthropogenic aerosols from industrial areas."

l.24 change to 'aerosol properties to be'

–Changed in the manuscript.

l.25 an aim? and later remove ', for example,'

–Removed from the manuscript.

l.27 consider changing 'things and situations' to 'data'

–Changed to "data".

l.30 change to 'The ANN contain tens'

–Corrected in the manuscript.

p.4 l.4-5 change to 'In this paper, an in-house developed ANN algorithm for aerosol typing is introduced.'

–Changed to "In this article, an in-house-developed ANN algorithm for aerosol typing is introduced."

l.9 change to 'are used at first to compute'

–Changed in the manuscript.

l.10 start in new line: The ability...

–New paragraph added.

l.12 coma before 'including'

–Corrected in the manuscript.

l.13 change to 'of different aerosols was' later on 'capable of reproducing the observed aerosol properties, and'

–Corrected in the manuscript.

l.15 change to 'needed doe to sparse observational data sets'

–Changed in the manuscript.

l.16 change to 'The aerosol model was constructed to'

–Changed in the manuscript.

l.27 remove 'in this article

–Removed from the manuscript.

l.31 chemical composition <- give hint on which parameters

–The following sentence was added to the manuscript: "The chemical composition of each pure aerosol type was picked up from OPAC (Optical Properties of Aerosols and Clouds) software package (Hess et al. 1998)."

l.32 certain limits <- specify in brackets what limits you mean

–The following was added to the manuscript: "[. . .] certain limits (the limits are detailed in Table 2 and refer to particle number density mixing ratios)".

l.32 remove 'as much as possible'

–Removed from the manuscript.

p.5.1.4 remove 'average'

–Removed from the manuscript.

l.14-15 change to 'In the aerosol model, particles ... ratios to simulate the aerosol anisotropy'

–Changed to "In the aerosol model, particles were considered as spheroids with different axis ratios to simulate the aerosols anisotropy (Table 2)."

l. 17 microscopical properties <- give hint on which properties

–Changed in the manuscript to "[. . .] microphysical properties (i.e. mode radius, width of the log-normal distribution, number density, density and mass concentration) of each component".

l.18 change to 'in certain limits the critical component (in brackets give hint what you mean!)'

–Changed to "[. . .] in certain limits the critical component (i.e. its number density mixing ratio)".

l.24 change to 'For each'

–Changed in the manuscript.

p.6 l.1 change to 'for the j components of the aerosol mixture, respectively.'

–Changed in the manuscript.

l.4 change to 'The size ... is mono-modal, log-normal and given by ..... where sigma_p, sigma_j sigma is ... and r is ... ' (note: in Eg.5 is r not r_j ?)

–Equation 5 was corrected. The sentence was changed to "The aerosol size distribution (n (r)) as a function of aerosol radius (r) assuming mono-modal log-normal distribution is given by"

l.13 change to 'Use of the calculated microphysical properties with the T-matrix code (provide reference), the effective cross-section for the particle scattering C_sca and extinction C_ext, as well as ...'

–Changed in the manuscript. For the T-matrix code the following reference was added: Mishchenko, M. I., and L. D. Travis (1994), T-matrix computations of light scattering by large spheroidal particles, Opt. Commun., 109, 16–21.

l.14-15 and l.17-18 change to '... alpha is determined from Eq.6 and ... beta from Eq. 7, where F11 is...'

–Changed in the manuscript.

l.7-11 move to the end of l.20, then correct it as follows: The integration domain (R_min:R_max), for which the effective radius r?, r_p?, the extinction coefficient alpha and scattering coefficient sigma? sigma_p? are calculated, covers medium size particles with radius between ... The radius was not increased further due to computing time limitation and model design limitations (i.e. ...). However, the latter limitation is not considered critical for the range of the lidar wavelengths.

–Changed in the manuscript.

l.10 correct and start in new line: 'The single scattering albedo omega is yield as the ratio of the ....

–Corrected and changed in the manuscript.

p.7 l.5 change to 'The algorithm is iterated for each'

–Changed in the manuscript.

l.6 it is confusing -> what domain you mean here?

–To clarify the following sentence was added to the manuscript: "The domain represents the range in which the parameters are varied (e.g. the domain for the wavelength is [350, 500, 1000 nm]; the domain for RH is [50%, 70%, 80%, 90%]; the domain for the number density mixing ratios for each component of each pure aerosol type is listed in Table 2)."

l.11 in brackets change to '(above 90%)'

–Changed in the manuscript.

l.10 change to 'considered as too rarely present in the ambient atmosphere.'

–Changed in the manuscript.

l.12 change to 'Thus, within ... is considered linear for '

–Changed in the manuscript.

l.15 rather: 'This is partly due to the limitations of the model itself and partly due to the various uncertainties associated to the measurement ...'

–Changed in the manuscript.

l.16 explain what you mean by linearity here.

–Changed to "(e.g. biases, calibration)" (linearity was removed).

l.17 change to 'applied in pre-processing to correct' later on change to 'as close'

–Changed and corrected in the manuscript.

l.17-21 this is not very clear, do you mean that 'To reproduce lidar measurement, for each simulated parameter an associated relative error (deltax) is assumed and the possible variable range is then computed as [x_med-deltax:x_med+deltax]', were x_med is the mean? median? value within the selected layer boundaries? But then why this is applied to syntetic data and true measurement? So you add additional error to the true data? what you mean by 'certain step' in finesse? is the deltax relative or absolute error and finally which is assumed, both?

[Figure]

–The paragraph was changed to: "Optical parameters calculated from lidar measurements are reported in the EARLINET database as the mean value (xmed) and associated uncertainty (absolute error, $\Delta$x). Optical parameters calculated from synthetic data do not carry this uncertainty, therefore a fix relative error was considered which was multiplied with the value to obtain the absolute error (uncertainty). For the actual retrieval of aerosol type was considered that any value between (xmed uncertainty) and xmed +uncertainty) was possible, therefore the algorithm was applied for all these values with a certain step (i.e. the finesse). The output is a "bundle" of possible aerosol types, whit a dimension equal to the finesse. A compromise should be made between the finesse and computing time."

l.23 start in new line: Based on the values ...

–A new paragraph was added.

l.25 rather it should be: 'extinction coefficient derived with Raman method, mainly due to noisy Raman lidar signals.'

–Changed in the manuscript.

l.24,27, be specific, in brackets give uncertainties of ext, bscat and depol that you mean!

–The paragraph was changed to "Based on the values reported in the literature (e.g. Ansmann et al. 2002, Freudenthaler et al. 2009), a large uncertainty is associated with the extinction coefficient derived with the Raman method, mainly due to noisy Raman lidar signals (i.e. the relative error reported in the lidar measurements is 30% –150% and the fixed relative error considered in the synthetic data is 50%). Particle depolarization is very sensitive to the calibration, both for the raw signals of the two channels and for the backscattering. Thus, the values for particle depolarization have also a significant uncertainty (i.e. the relative error reported in the lidar measurements is 2%– 50% and the fixed relative error considered in the synthetic data is 30%). The

backscattering coefficient calculated from the combination of Raman-elastic channels is less sensitive (i.e. the relative error reported in the lidar measurements is 10%– 50% and the fixed relative error considered in the synthetic data is 20%)."

l.27-28 change to 'in the case'

–Corrected in the manuscript.

l.28 change to 'Lidars, were the'

–Changed in the manuscript.

l.29 change to 'cross-talk... is still'

–Changed in the manuscript.

l.31 change to 'inclusive to mimic ...parameters.'

–Changed in the manuscript.

p.8 l. 2 add come before 'as long as'

–Corrected in the manuscript.

l.5-6 remove sentence 'There was.. mixtures' and then change to 'The mixtures composed of only two pure types were considered not sufficient.'

–Sentence removed. Changed in the manuscript.

l.10-11 change to 'of possible mixtures ... three pure aerosols (35 mixtures), only ... (9 mixtures, see Table 4).'

–Changed in the manuscript.

l.12 you mean: the time-performance of the algorithm and the minimum number of output aerosol types considered significant in atmosphere.

–Changed in the manuscript as indicated by the reviewer.

l.13-14 correct and change to 'the extinction Angstrom exponent (k_ext) and the backscattering Angstrom exponent (k_bsca), also referred to as color ratio'

–Changed in the manuscript.

l.21 titla of 2.2: The ANN architecture ?

–Title of the subsection changed to "The architecture Artificial Neural Networks"

l.22-26 belongs to Introduction! -> start the section with: The ANNs can be calibrated or...

–The text on lines 22–24 was removed from section 2.2 and integrated in the introduction.

l.2 change to 'elements of identical parameters (e.g. ...) with an associated ...'

–Changed in the manuscript.

l.4 start as new paragraph: No significant ...

–New paragraph added.

l.5-6 unclear sentence: 'The TanhAxon ... -1 and 1.' you mean: a bias-offset B ? each neuron x_i ? in the layer omega_i ?

–Changed to: "The TanhAxon applies a bias and a hyperbolic tangent function (i.e. tanh) to each neuron in the layer and replaces a part of the tanh by a line having a slope beta. The values of each neuron are forced to be in the interval -1 and 1. For TanhAxon the activation function is defined as".

l.9 remove 'all'

–Removed from manuscript.

l.9 start in the new line with: Supervised ...

–New paragraph added.

[Figure]

l.13 change to ' training approach... ANN: the input... and the output compared with the [specify: true? or initially assumed? or pure?] aerosol type, in order...'

–Changed in the manuscript to "[. . .] known aerosol type from synthetic database."

l.15 specify: the minimum classification errors?

–The following details were provided: "The minimum classification errors (i.e. confidence of classification) were 75% for more than 80% of the measured data and 75% for more than 90% of the synthetic data."

l.17 clarify: the standard gradient was not tested at all?

–The standard step gradient has been tested only during the first tests and then removed due to low performances, while other learning rules have been intensively tested during the entire network build up process.

l.18 change to 'It provides'

–Changed in the manuscript.

l.21 change to The ConjugateGradient ... has no parameters ... (such as learning ...) and'

–Changed to "The Conjugate Gradient has no parameters to be adjusted (e.g. learning rates, momentum parameter) and is faster and more accurate with respect to the standard backpropagation."

l.22 change to 'algorithm'

–Changed in the manuscript.

l.24 change to 'function, were found inadequate for aerosol typing purpose.'

–Changed in the manuscript.

l.25 change to 'but its active'

–Changed in the manuscript.

l.26 start as new paragraph: The cross-validation ...

–New paragraph started

l.27 please make it more specific here what you mean by 'when error begins to increase'? is it at the beginning zero? or has some initial constant value? how is it defined? assumed?

–Cross validation is intended to test if the training process is good enough. Even if the training errors are decreasing, in the sense of better learning, actually the neural network is somehow "remembering" the right response not improving anymore. In order to overpass this, a small set of data (not shown before) are used to test the learning process. In the test case the errors can increase so the learning process is not improving and should be stopped. Thus, the cross validation is used to stop the learning process, when it is reached the maximum potential. The initial matrix starts from zero.

l.29 be more specific, what you mean by the exact type?

–Changed to "known aerosol type from the synthetic database".

l.30-31 change to In total 68 ANN structures ... in order to compromise between the minimum ...

–Changed to "In total 68 ANN structures have been explored, starting from the simplest (reduced number of hidden layers) to the complex ones, in order to compromise between the minimum possible time of training and testing, and avoiding "saturation" effects"

l.31 specify what is meant here by saturation effects?

–The structure and complexity of ANN represents a compromise. If the architecture is more complex, the classification can be more accurate, but at the costs of drastically

increasing the computation time. While an ANN cannot provide a good classification if is not complex enough.

l. 32-33 rather change to 'Examples of 6 pure, 7 double-component mixtures and 2 triple-component mixtures obtained within the 68 explored ANN ...

–Changed to "Examples of 6 pure, 7 double-component mixtures, and 2 triple-component mixtures obtained within the 68 explored ANN are presented in Table 4."

p.10 l.2-5 state clearly if for the training, tasting and validation the same classification error <25% is used.

–The training, testing and validation classification errors are below 25%. Changed to "The training is performed iteratively until the testing and validation classification errors are below 25% (Fig. 2)."

l.7-8 should this not read 'the Jordan-Elman with 6 or 8 hidden layers and the Generalized Feedforward with 10 hidden layers'?

–Changed to "[. . .] the Jordan-Elman with 6 or 8 hidden layers, and the Generalized Feedforward with 10 hidden layers (Table 3)."

l.9 what you mean by response with higher confidence? the classification error?

–The ANN response is represented by the probability of having one of the aerosols types (e.g. 0.99), which define the confidence of the response. The sentence was changed to "The selected types of ANNs classify the aerosols based on the response with higher confidence (i.e. the probability of having one of the aerosols types)".

l. 10 define the best weights

–The best weights are reached when the training process is finished, so we have the lower classification errors. The sentence was to "The ANNs have been trained using 3500 samples for each aerosol type and successive training sessions until the best weights are reached (i.e. the classification process is ended, and the classification

errors are low)."

l.11 the title of 2.3 The typing algorithm?

–Changed in the manuscript.

l.12 should not be 'the uncertainty threshold'?

–Changed in the manuscript.

l.12-13 change to 'parameter, a bundle of imputs ... was generated.'

–Changed in the manuscript.

l.14 remove: 'then'

–Corrected in the manuscript.

l.15 remove: 'and'

–Corrected in manuscript.

l.16 start at new line: For the NATALI aerosol typing algorithm, the input ...

–New paragraph added. Changed to "The inputs parameters for NATALI are typical data products [...]"

l.17 optionally?

–Corrected in manuscript.

l.18 remove: all

–Removed from the manuscript.

l.21 good quality <- specify in brackets the uncertainty threshold

–The following details were added "(uncertainty of the aerosol extinction coefficient $\leq$50%, uncertainty of the aerosol backscatter coefficient $\leq$20%, uncertainty of the particle linear depolarization ration ≤30%)".

l.24 high uncertainty <- specify in brackets the uncertainty threshold range

–The following clarifications were added "(uncertainty of the aerosol extinction coefficient >50%, uncertainty of the aerosol backscatter coefficient >20%, uncertainty of the particle linear depolarization ration >30%)".

l.25 change to 'available, the'

–Changed in the manuscript.

l.26 change to 'meaning again that the'

–Changed in the manuscript.

l.27 change to 'retrieved for 5 outputs (...), whereby if... from Dust nor Continental ..., and therefore is ...'

–Changed in the manuscript.

l.30-31 change to 'The three ANNs (Table 3) were developed ...schemes (Table 4) to increase the confidence of the aerosol typing. A voting procedure selects the most...

–Changed in the manuscript.

p.11 l.1 the title of 2.4 The NATALI Code?

–Changed to "The NATALI code".

l.2 change to 'The ... developed in the Phyton programing environment is buitl on ...

–Changed in the manuscript.

l.6-7 change to 'calculates within each layer the mean intensive optical parameters (name them in brackets) with uncertainties (Fig.3).'

–Changed to "[. . .] calculates within each layer the mean intensive optical parameters

(i.e. Angstrom exponent, color indexes color ratios, lidar ratios, particle linear depolarization ratio) and their associated uncertainty)".

l.8 should be: The layer boundaries

–Corrected in the manuscript.

l.8 please clarify what is used boundary layer detection: the gradient (1st derivative of signal) or as in l.9 the inflection point (2nd derivative of signal)

–The gradient method is applied, i.e. locating the maxima and the minima of the 1st derivative, which is equivalent with locating the inflexion points of the 2nd derivative.

l.8 is the range&time resolution of 1064 signal the same as the resolution of the optical EARLINET profiles?

–The algorithm works the same regardless of the resolution of the profile, only the uncertainty of the location of the layers differs. The algorithm uses the resolution reported in EARLINET.

l.9 specify what parameters for SG-filter are used, are they applied the same for all profiles?

–Changed to "The inflexion points of the second derivative of the profile data, computed with the Savitzky-Golay filter, give the top and the bottom of the layers. The window size of the cubic Savitzky-Golay filter, which be modified by the user, has a default value of 700 m. The filter was applied twice to obtain the second derivative. A signal-to-noise ratio filter is applied at this point, making sure the said ratio is at least 5. The layer boundaries are moved towards the median height until the SNR criteria is met; if the criteria cannot be satisfied with a layer height greater than 300m, the layer is discarded. Gross or fine structure of the aerosol layers is revealed by a higher or lower value of the adjustable smoothing parameter (FINESSE). The layers with thickness of more than 300 m are considered, whereby the intensive optical properties and their uncertainties are computed for the middle of each layer in the range of at least 200 m

thickness, to exclude the margins likely affected by the smoothing."

l.11-14 From own experience, layers of thickness < 300m can have higher SNR than layers > 300m, as SNR depends on aerosol optical depth, which even for very thin layers can also be very high! I suggest to revise the fragment as follows: The layers with thickness of more than 300 m are considered, whereby the intensive optical properties and their uncertainties are computed for the middle of each layer in the range of at least 200m thickness, to exclude the margins likely affected by the smoothing.

–Changed in the manuscript.

l.15-16 The first 2 sentences are repetitions, pls remove it, and start directly with: Several filters ... pass the following criteria ...

–Changed in the manuscript.

l.22 change to 'the input model generates an adjustable number N of values x with uncertainties deltax in the range of [x-deltax:x_deltax]'.

–Changed in the manuscript.

l.24 start in new line: The typing module ...

–A new paragraph was added.

l.25 change to 'In the case that the depolarization ratio is'

–Corrected in the manuscript.

l.28 remove 'some'

–Removed from the manuscript.

l.29-32 change to ', and returns only the ... type (Volcanic overlaps completely (...) ...polluted type and cannot be retrieved in low-resolution.'

–Changed in the manuscript.

l.32 remove as irrelevant sentence: Thus...

–Removed from the manuscript.

l.33 change to 'telegrams contain the identification of the data sets for which typing is performed and provides for each identified layer following parameters:'

–Changed to "The csv files and the telegrams contain the identification of the data sets for which typing is performed and provides for each identified layer the following parameters:"

p.12 l.3-7 start each parameter with 'the'

–Corrected in the manuscript.

l.10 change to 'The NATALI code'

–Changed in the manuscript.

l.10-12 check the names of the python routines there is problem with the font.

–The name of the routines/codes are written in LaTex in the same font as the equations to differentiate then from the rest of the text. We also made correction for the names.

l.12 should be: The three

–Corrected in the manuscript.

l.13 rather module routines/codes than methods!

–Changed in the manuscript.

l.15-18 should be Firstly, ... Secondly, ... Thirdly, ...

–Corrected in the manuscript.

l.22 change to 'by the developed aerosol model'

–Changed in the manuscript.

l.25 change to 'As shown in'

–Corrected in the manuscript.

l.26 move the '(i.e. ...) to the end of the sentence.

–Moved at the end of the sentence.

l.27-29 could you comment on what could be the reason for the given discrepancies?

–This explanation was added: "The reasons for these discrepancies are many-fold. In some cases, values reported in the literature have high uncertainties because of natural variability, improper calibration and retrieval. The aerosol model has also some limitations, e.g. due to spheroidal model and mono-modal log-normal distribution considered."

p.13 l.3 OPAC hygroscopicity was NOT available?

–Corrected in the manuscript.

l.3-4 change to 'However, the changes in OPAC are not expected to produce ...'

–Changed in the manuscript.

l.6-8 chnage to 'In the Figure 4 comparisons ... by Gross et al. (2013) are provided.

–Changed in the manuscript.

l.8-13 Based on the Airborne... properties measured during campaigns in ... EU-CAARI), Gross et al. (2013) developed an aerosol classification scheme for six aerosol types ...'

–Changed to: "Based on the Airborne High Spectral Resolution Lidar (HSRL) data and in situ measurements of aerosol microphysical and optical properties collected during a series of measurements campaigns in 1998 (Lindenberg Aerosol Characterization Experiment, LACE), 2006 (The Saharan Mineral Dust Experiment, Morocco, SAMUM-1), and 2008 (The Saharan Mineral Dust Experiment, Cape Verde Islands, SAMUM-2

and European integrated project on Aerosol Cloud Climate, EUCAARI), Groß et al. 2013 developed an aerosol classification scheme for six aerosol types and aerosol mixtures (i.e. Saharan mineral dust, Saharan dust mixtures, Canadian biomass burning aerosol, African biomass burning mixture, anthropogenic pollution aerosol, and marine aerosol)."

l.15 should be: observations at 532 nm from ... (Fig.4 a and d), especially for smoke...

–Changed in the manuscript.

l.17 should be: therefore were not compared.

–Changed in the manuscript.

l.18-20 should be: Clusters are identified both in synthetic and observational data, which means ... one wavelength can be sufficient for the ANN training.

–Changed in the manuscript.

l.18-20 I am interested if you did actually try to train the ANNs with the single wavelength extinction, backscattering and depolarization?

–No. Different types of ANNs have to be selected for this, work in progress.

l.23 either add after 'measurement campaigns' comment in brackets: (listed in caption of Fig.5)' or list the campaigns in the text and leave them out in the Fig.5 caption, what will be consistent with Fig.4

–Changed to Wandinger et al. (2016) provided a synthesis of ground-based observations of lidar ratio and particle linear depolarization at 355 nm for different aerosol types (i.e. dust, smoke, pollution, marine, aerosol, volcanic ash) and mixtures, collected during a series of measurements campaigns, i.e. at Cape Verde (Groß et al. 2011), at EARLINET stations of Leipzig and Munich (Groß et al. 2012), in the Amazon Basin (Baars et al. 2012), and over the North Atlantic (Kanitz et al. 2013).

l.29 it is difficult to see overlapping of smoke and continental polluted

–We believe that this overlapping is shown more clearly in the new Fig. 6.

l.29-31 I would revise this fragment as follows: Smoke and continental pollution almost completely overlap (Fig.6a), which is consisted with measurements reported in literature as in Table 6. This makes the typing challenging. The importance of particle depolarization shown relatively recently (e.g. =Freudenthaler et al., 2019), can improve the aerosol typing (Fig.6b).

–Changed to "Smoke and continental pollution almost completely overlap (Fig. 6a), which is consisted with measurements reported in literature (Table 6). This makes the typing challenging. The importance of particle depolarization shown relatively recently (e.g. Freudenthaler et al. 2009), can improve the aerosol typing (Fig. 6b)."

l.33 should be: calibration, recently addressed by e.g. McCullough ...

–Changed in the manuscript.

l.34 should be: the depolarization ratio quality criteria.

–Changed in the manuscript.

l.34-35 rephrase this sentence to positive: However, even without particle depolarization information, the low-resolution typing can identify the aerosol predominant types in a mixture.

–Changed in the manuscript.

p.14 l.9 change 'Also,' to '; continental ...'

–Changed in the manuscript.

l.10 change 'while' to '.Note that 33% ...'

–Changed in the manuscript.

l.12 add at the end of 1st sentence: (Fig.7b).

–Added in the manuscript.

l.16 start in new line: A3H and A3L ...

–New paragraph added.

l.16 most or best performing?

–Changed to "best".

l.18-19 last sentence -> pls clarify, I do not see this from provided Figs!

–This situation is not visible in the figures because the voting procedure rejects results bellow a certain confidence level.

l.21 change 'for example,' to 'olny'

–Changed on the manuscript.

l.21 I think there should be dust continental instead of marine mineral for high-resolution typing

–Corrected in the manuscript.

l.22 I think there should be continental smoke and mixed smoke for low-resolution typing

–Corrected in the manuscript.

l.23 consider adding sentence: For costal polluted, the relative humidity increase results in an increase of typing performance.

–The sentence was added to the manuscript.

l.25 I suggest to separate the CALIPSO and INOE data sets, therefore I propose to rewrite this as: Observational data from EARLINET Data Base (www.earlinet.org), related to the CALIPSO (...) overpasses over different EARLINET observational sites, were compared with ... model. The EARLINET-CALIPSO Data base (Pappalardo et al. 2010), covers the data of 2000-2018 ??? and includes...

–Changed to: "Observational data from EARLINET Data Base (www.earlinet.org), related to the CALIPSO (Cloud-aerosol Lidar and Infrared Pathfinder Satellite Observation) overpasses over different EARLINET observational sites, were compared with the synthetic data obtained from the aerosol model. The EARLINET-CALIPSO Data base (Pappalardo et al. 2010a), covers the data of 2000–2018 and includes a total of 718 cases and 21 aerosol and cloud types."

l.29 should be: all of the

–Corrected in the manuscript.

l.31-32 change to: '... was added, assuming values reported in literature as typical for the corresponding aerosol type. This way, 105 cases ... were obtained.'

–Changed to "To increase the number of cases, the particle depolarization was added assuming values reported in literature as typical for the corresponding aerosol type. This way, 105 cases containing all needed parameters were obtained."

p.15 l.1-2 change to: 'The cases for which all parameters were within 20% of relative error were selected (63 cases), whereby 57 corresponded to known aerosol types.'

–Changed in the manuscript.

l.3 add sentence: 'Additionally, profiles available at the EARLINET site in Bucharest/Magurele, established by the Romanian National Institute for Research and Development of Optoelectronics (INOE), were used to increase the validation measurements sample.'

–Sentence added to the manuscript.

l.4-5 change to: 'About 44.6% of measurements were conducted at nighttime (thus,

include the Rama-derived extinction coefficient profiles).

–Changed in the manuscript.

l.5 871 profiles or layers?

–Changed to "layers".

l.6 remove: profiles

–Removed from the manuscript.

l.6 instead of maximum of 1h, give range of temporal averaging (e.g. 30min-1h)

–Changed to "profiles averaged over one hour".

l.6 significant aerosol load ?

–Changed to "[. . .] significant loads (i.e. layers for which the uncertainty of the retrieved optical parameters is below the limits accepted by the algorithm)".

l.7 change 'could be' to 'were' later on should be 'higher than 20%'

–Changed in the manuscript.

l.10-11 in effect this are 311 and 182 layers?

–Changed to: "Thresholds were then used to estimate at first glance the type of aerosol which resulted in a dataset with 311 layers were acceptable by the algorithm, out of which for only 182 layers auxiliary data was available. Auxiliary data were used to compare the results of the typing."

l.11 start last sentence is new line as: The time-series of lidar measurements (532 nm vol.depol.ratio and ... range corrected signals) were used to identify the aerosol layers.

–Changed in the manuscript.

l.13 start last sentence in new line : The identification...

–New paragraph added.

l.14-16 change to: 'source was assumed at the region ... was closest to the ground, providing guidance ... The rainfall along trajectory was used a indicator for likely wet deposition.'

–Changed to: "The source was assumed to originate at the region where the trajectory was closest to the ground, providing guidance for identifying possible emission sources. The rainfall along trajectory was used as an indicator for likely wet deposition."

l.16 meteo parameters or fields?

–Changed to "fields".

l.17-18 should be: was based on ... reanalyses (Kalnay et al., 1996) to confirm the aerosol trajectories ...

–Changed to: "A synoptic diagnosis of the main meteorological filed (e.g. pressure, geopotential height, temperature, relative humidity, wind), based on NCEP/NCAR Re-analysis (Kalnay et al. 1999), was used as to confirm the aerosols trajectories and also to determine the type of atmospheric circulation, weather regimes and weather phenomena along the trajectories."

l.21-23 be specific: the mixtures of three components, incorrect calibration or inappropriate estimation of aerosol type. On the other hand the sparse observational data lead to ...

–Changed to:" The large spread of the measured parameters is caused by the mixtures of three components, incorrect calibration or inappropriate estimation of aerosol type. On the other hand, the sparse observational data lead to apparently incomplete clusters."

l.25 make new sentence: Low values ... lidar ratio are seen for several cases of ...

–Changed to: "Low values are observed in the Angstrom exponent for several cases of

dust polluted and smoke categories, as well as low values for the 532 nm lidar ratio are seen for several cases of continental and continental dust indicating a small portion of marine particles."

l.26 should be: This is most likely ...

–Corrected in the manuscript.

l.27 coma before 'and thus'

–Corrected in the manuscript.

l.29 change 'could not' to 'failed to'

–Changed in the manuscript.

l.29 consider adding information: Actually, in general the agreement of classification of the simulated data and the real observations is very good, given all limitations discussed.

–Added in the manuscript.

p.16 l.2-4 rather unclear, especially the prerequisite hypothesis

–We have tried to make this sentence as clear as possible.

l.6 strong or crucial?

–Changed to "crucial".

l.7 should be: the depolarization methodology is still maturing and only a few lidar stations ...

–Changed in the manuscript.

l.15 should be: for the input data gave realistic retrieval of aerosol ...

–Changed in the manuscript.

l.21 I reckon you should repeat very valuable information in the section: Software Availability next to the Acknowledgements!

–A new section on "Code Availability" was added after Conclusions.

l.22 easy to use or: The NATALI software is user-friendly; a user-guide is provided.

–Changed in the manuscript.

l.23 should be: the limitations of the algorithm, i.e. the results are ... on the quality of the input data

–Changed in the manuscript.

l.27 should be: of less that

–Corrected in the manuscript.

l.28 rather: of the cases for medium and high

–Changed in the manuscript.

l.33 rather: for example, it can be applied to test ... data, identify ...

–Changed in the manuscript.

l.33 could you please comment whether you did actually tested this specific QC procedure? is there any conference paper on that you could cite?

–A citation was added to a conference paper by Nicolae et al. (2018).

p.17 l.2-3 comment on expected results for data sets of more than 3beta, 2 alpha and 1sigma, e.g. adding polarization at 355 and/or 1064 nm; adding lidar-derived RH profiles? How you expect this improves the results?

–Changed to: "More complex datasets (e.g. availability of LPDR at 355 and/or 1064 nm) will not produce improvements with the current software because ANNs are specifically trained for 3beta + 2alpha + 1delta datasets. However, the ANNs can be trained

with more complete datasets, which potentially can lead to better scores, especially in the case of mixtures."

References

p.21 l.17 should be Janicka; p.25 l.10 should be Muller, D.; p.25 l.13 remove 2x .and

–Corrected in the manuscript.

Equations

Eg.9 more elegant formulation: LR=4pi/omega*F11(180)

–Changed in the manuscript.

Eg. 13 please do not use beta and omega in this equation, as they both have already associated physical quantities! the current beta-bias-offset can be denoted as B. What is the omega here? as there is x_iËĘlin is this not LinearThanAxon?

–Corrected in the manuscript. This is the bias vector.

Figures

Fig.1&2&3 improve captions, specifically add info that schemes are used for the NATALI code.

–Changes in the manuscript to: "Fig. 1. The generation chain of the synthetic data for the NATALI algorithm."; "Fig. 2. Artificial Neural Network logical scheme for the NATALI algorithm.", and "Schematics of the NATALI algorithm for aerosol typing."

Fig.3 I suggest to change the description to: Mean intensive optical parameters within the layer, Mean uncertainty of intensive optical parameters within the layer, Scrambled seta of mean intensive optical parameters +- uncertainty

–Changed in Figure 3.

Fig.4 indicate in upper sub-figs. what wavelengths they depict; for consistency in legend of lower sub-figs give aerosol types in sequence: dust, dust polluted, smoke, continental polluted, marine

–Figure 4 was changed to incorporate the suggestion from the reviewer.

Fig.5 indicate for which wavelength is the upper figure! for comparisons the same wavelengths of the observations and synthetic data must be used. Caption is lengthy, the description of the campaigns should go to the text.

–Figure 5b was replaced using 350 nm and the length of the caption was reduced ("Lidar ratio versus particle linear depolarization ratio. (a) Synthesis of ground-based observations and simulations adapted from Wandinger et al. (2016) (their Fig. 1). Filled stars represent simulations using the components of Aerosol CCI and variations with different refractive index and shape distribution (open stars). (b) Synthetic data from the NATALI aerosol model.").

Figs.6-9 for consistency, order the aerosol types as in AH definition in Table 4 and use THE SAME COLOR SCALE for each aerosol type.

–Figure 6 is using the same color scheme as in Figure 9. The aerosols types are defined in the same order as in Table 4 (AH column) in Figs. 6–9.

Fig.6 upper figure should be CRvsLR not ARvsDR! In caption add info that this is NATALI classification.

–Changed from ARvsDR to CRvsLR in Fig. 6a. Caption was also changed ("[...] using the NATALI classification").

Fig.7&8 change Marine/CC to marine! Plot confidence-levels? vote-number? in grayscale (white being 0-20%, black 80-100%). Note that low-resolution typing for 7 pure categories is excellent. Note that high and low resolution results are better of any info on RH is available. Give more explanation on how to understand figs in the captions. Again, add info that this is for NATALI code.

–"Marine/CC" was changed to "marine" and NATALI code was added to figure caption. Figure 7 caption was changed to:" Performances of ANNs for (a) high-resolution typing, and (b) low-resolution typing for each ANN (i.e. A1H, A2H, A3H) and the combine results (Vote) of the three ANNs. The intervals of ANNs confidence levels are shaded according to the scale." Figure 8 caption was changed to "Performances of the ANNs for different relative humidity values (50%, 70%, 80%) for (a) high-resolution typing, and (b) low-resolution typing. The intervals of ANNs confidence levels are shaded according to the scale."

Fig.9 Add info that INOE site is also EARLINET site. Add over the left column info synthetic-data and over the right one observational-data

–We have added "synthetic data" on the left column and "observational data" over the right column. Figure caption was changed to: "Results of the aerosol typing from NATALI aerosol model (synthetic data) and observations (observational data, EARLINET-CALIPSO database and additional datasets collected at the EARLINET station in Bucharest). (a) and (b) lidar ratio and particle depolarization (VIS), (c) and (d) Angstrom exponent and particle depolarization (VIS), and (e) and (f) lidar ratio (VIS) and lidar ratio (UV)."

Tables

Table 1 change big to large; change right column title to Particle characteristics, for marine -> should not be non-absorbing? continental polluted and smoke are identical but smoke can be aspherical.

–Changed in the manuscript.

Table 2 what is in the 3th column, whet is this critical component proportion? is has no units? you mean axis-ratio or aspect-ratio? Is this table defined by results reported in literature?

–The header of the 3rd column was changed to "Range variation of the number density

mixing ratios for aerosol components (limits are consistent with OPAC and literature)".
"Axis ratio" was changed to "aspect ratio".

Table 3 check font of than-axons

–Corrected in the manuscript.

Table 4 specify whether aerosol types are retrieved directly by the NATALI or measurements? Note there is no italic font! why categories in Arial-font are given in low resolution if they are not really retrieved?

–Table caption was replaced with "Correspondence between the aerosol types defined in the algorithm, as they can be retrieved by NATALI in high resolution and low resolution."

Table 5 AE and CR have the same values? is the AE, CR of 6 realistic? the Rayleigy being 4 should not be the limit? Which CI is correct? Is the lower CI not LR? Please check carefully this table for typos!

–The errors in the table were corrected.

Table 6 consider adding reference for Continental polluted/industrial: Raman lidar derived AE550/350 of 1.17-1.19 is reported in Remote Sensing 2017, 9(11), 1199; doi:10.3390/rs9111199.

–Reference and the results were added to Table 6.